# Non-convex Distributionally Robust Optimization: Non-asymptotic Analysis

**Jikai Jin**[1,*]  **Bohang Zhang**[2,*]  **Haiyang Wang**[3]  **Liwei Wang**[2,3,4,†]

[1]School of Mathematical Sciences, Peking University
[2]Key Laboratory of Machine Perception, MOE, School of EECS, Peking University
[3]Center of Data Science, Peking University    [4]Institute for Artificial Intelligence, Peking Unviersity
{jkjin,zhangbohang}@pku.edu.cn, wanghaiyang6@stu.pku.edu.cn, wanglw@cis.pku.edu.cn

## Abstract

Distributionally robust optimization (DRO) is a widely-used approach to learn models that are robust against distribution shift. Compared with the standard optimization setting, the objective function in DRO is more difficult to optimize, and most of the existing theoretical results make strong assumptions on the loss function. In this work we bridge the gap by studying DRO algorithms for general smooth non-convex losses. By carefully exploiting the specific form of the DRO objective, we are able to provide non-asymptotic convergence guarantees even though the objective function is possibly non-convex, non-smooth and has unbounded gradient noise. In particular, we prove that a special algorithm called the mini-batch normalized gradient descent with momentum, can find an $\epsilon$-first-order stationary point within $\mathcal{O}(\epsilon^{-4})$ gradient complexity. We also discuss the conditional value-at-risk (CVaR) setting, where we propose a penalized DRO objective based on a smoothed version of the CVaR that allows us to obtain a similar convergence guarantee. We finally verify our theoretical results in a number of tasks and find that the proposed algorithm can consistently achieve prominent acceleration.

## 1 Introduction

For a classical machine learning problem, the goal is typically to train a model over a training set that achieves good performance on a test set, where both the training set and the test set are drawn from the *same* distribution $P$. While such an assumption is reasonable and simple for theoretical analysis, it is often not the case in real applications. For example, this setting may be improper when there is a gap between training and test distribution (e.g. in domain adaptation tasks) [Zhang et al., 2021], when there is severe class imbalance in the training set [Sagawa et al., 2020], when fairness in minority groups is an important consideration [Hashimoto et al., 2018], or when the deployed model is exposed to adversarial attacks [Sinha et al., 2018].

Distributionally robust optimization (DRO), as a popular approach to deal with the above situations, has attracted great interest for the machine learning research communities in recent years. In contrast to classic machine learning problems, for DRO it is desired that the trained model still has good performance under distribution shift. Specifically, DRO proposes to minimize the worst-case loss over a set of probability distributions $Q$ around $P$. This can be formulated as the following constrained optimization problem [Rahimian and Mehrotra, 2019, Shapiro, 2017]:

$$\text{minimize}_{x \in \mathcal{X}} \quad \Psi(x) := \sup_{Q \in \mathcal{U}(P)} \mathbb{E}_{\xi \sim Q} \left[ \ell(x; \xi) \right] \tag{1}$$

---

[*]Equal Contribution, alphabetical order.
[†]Corresponding author.

35th Conference on Neural Information Processing Systems (NeurIPS 2021).

where $x \in \mathcal{X}$ is the parameter to be optimized, $\xi$ is a sample randomly drawn from distribution $Q$, and $\ell(x; \xi)$ is the loss function so that $\mathbb{E}_{\xi \sim Q}[\ell(x; \xi)]$ is the expected loss over distribution $Q$. The DRO objective $\Psi(x)$ is therefore the worst-case loss when the distribution $P$ is shifted to $Q$. The set $\mathcal{U}(P)$ is called the uncertainty set and typically defined as

$$\mathcal{U}(P) := \{Q : d(Q, P) \leq \epsilon\} \tag{2}$$

where $d$ measures the distance between two probability distributions, and the positive number $\epsilon$ corresponds to the magnitude of the uncertainty set.

Instead of imposing a hard constrained uncertainty set, sometimes it is more preferred to use a soft penalty term, resulting in the penalized DRO problem [Sinha et al., 2018]:

$$\text{minimize}_{x \in \mathcal{X}} \quad \Psi(x) := \sup_{Q} \{\mathbb{E}_{\xi \sim Q}[\ell(x; \xi)] - \lambda d(Q, P)\} \tag{3}$$

where $\lambda > 0$ is the regularization coefficient.

There are many possible choices of $d$. A detailed discussion of different distance measures and their properties can be found in Rahimian and Mehrotra [2019]. In this paper we consider a general class of distances $d$ called the $\psi$-divergence, which is a popular choice in DRO literature [Namkoong and Duchi, 2016, Shapiro, 2017]. Specifically, for a non-negative convex function $\psi$ such that $\psi(1) = 0$ and two probability distributions $P, Q$ such that $Q$ is absolutely continuous w.r.t. $P$, the $\psi$-divergence between $Q$ and $P$ is defined as

$$d_\psi(Q, P) := \int \psi\left(\frac{\mathrm{d}Q}{\mathrm{d}P}\right) \mathrm{d}P.$$

which satisfies $d_\psi(Q, P) \geq 0$ and $d_\psi(Q, P) = 0$ if $Q = P$ a.s.

The main focus of this paper is to study efficient first-order optimization algorithms for DRO problem (3) for *non-convex* losses $\ell(x, \xi)$. While non-convex models (especially deep neural networks) have been extensively used in DRO setting (e.g. Sagawa et al. [2020]), theoretical analysis about the convergence speed is still lacking. Most previous works (e.g. Levy et al. [2020]) assume the loss $\ell(\cdot, \xi)$ is convex, and in this case (3) is equivalent to a convex optimization problem (see Section 2 for details). Recently some works provide convergence rates of algorithms for non-convex losses in certain special cases, e.g. the divergence measure $\psi$ is chosen as the conditional-value-at-risk (CVaR) and the loss function has some nice structural properties [Soma and Yoshida, 2020, Kalogerias, 2020]. Gürbüzbalaban et al. [2020] considered a more general setting but only proved an asymptotic convergence result for non-convex DRO.

Compared with these works, we provide the first *non-asymptotic* analysis of optimization algorithms for DRO with *general smooth non-convex* losses $\ell(x, \xi)$ and general $\psi$-divergence. In this setting, there are two major difficulties we must encounter: (i) the DRO objective $\Psi(x)$ is non-convex and can become arbitrarily *non-smooth*, causing standard techniques in smooth non-convex optimization fail to provide a good convergence guarantee; (ii) the noise of the stochastic gradient of $\Psi(x)$ can be arbitrarily large and unbounded even if we assume the gradient of the inner loss $\ell(x, \xi)$ has bounded variance. To tackle these challenges, we propose to optimize the DRO objective using *mini-batch normalized SGD with momentum*, and we are able to prove an $\mathcal{O}(\epsilon^{-4})$ complexity of this algorithm. The core technique here is to exploit the specific structure of $\Psi(x)$, which shows that (i) the DRO objective satisfies a generalized smoothness condition [Zhang et al., 2020a,b] and (ii) the variance of the stochastic gradient can be bounded by the true gradient. This motivates us to adopt the special algorithm that combines gradient normalization and momentum techniques into SGD, by which both non-smoothness and unbounded noise can be tackled, finally resulting in an $\mathcal{O}(\epsilon^{-4})$ complexity similar to standard smooth non-convex optimization.

The above analysis applies to a broad class of divergence functions $\psi$. We further discuss special cases when $\psi$ has additional properties. In particular, to handle the CVaR case (a non-differentiable loss), we propose a divergence function which is a smoothed variant of CVaR and is further Lipschitz. In this case we show that a convergence guarantee can be established using vanilla SGD, and an similar complexity bound holds.

We highlight that the algorithm and analysis in this paper are not limited to DRO setting, and are described in the context of a general class of optimization problem. Our analysis clearly demonstrates the effectiveness of gradient normalization and momentum techniques in optimizing ill-conditioned objective functions. We believe our result can shed light on why some popular optimizers, in particular Adam [Kingma and Ba, 2015], often exhibit superior performance in real applications.

**Contributions.** We summarize our main results and contributions below. Let $\psi^*$ be the conjugate function of $\psi$ (see Definition 2.3). For non-convex optimization problems, since obtaining the global minima is NP-hard in general, this paper adopts the commonly used (relaxed) criteria: to find an $\epsilon$-approximate first-order stationary point of the function $\Psi$ (see Definition 2.5). We measure the complexity of optimization algorithms by the number of computations of the stochastic gradient $\nabla\ell(x,\xi)$ to reach an $\epsilon$-stationary point.

- Assuming that $\psi^*$ is smooth and the loss $\ell$ is Lipschitz and smooth (possibly non-convex or unbounded), we show in Section 3.2 that the mini-batch normalized momentum algorithm (cf. Algorithm 1) has a complexity of $\mathcal{O}(\epsilon^{-4})$.

- Assuming that $\psi^*$ is further Lipschitz, in Section 3.4 we prove that vinilla SGD suffices to achieve the $\mathcal{O}(\epsilon^{-4})$ complexity. As a special case, we propose a new divergence which is a smoothed approximation of CVaR.

- We conduct experiments to verify our theoretical results. We observe that our proposed methods significantly accelerate the optimization process, and also demonstrates superior test performance.

## 1.1 Related work

**Constrained DRO and Penalized DRO.** There are two existing formulations of the DRO problem: the constrained DRO and the penalized DRO. The constrained DRO formulation (1) has been studied in a number of works [Namkoong and Duchi, 2016, Shapiro, 2017, Duchi and Namkoong, 2018], while other works consider the penalty-based formulation (3) [Sinha et al., 2018, Levy et al., 2020]. From a Lagrangian perspective, the two formulations are equivalent; however, the dual objective of the constrained formulation is sometimes hard to solve as pointed out in [Namkoong and Duchi, 2016, Duchi and Namkoong, 2018]. In this paper we focus on the penalty-based version and provide the first non-asymptotic analysis in the non-convex setting. Moreover, we do not make the assumption that the loss is bounded, as assumed in Levy et al. [2020] in the convex setting.

**DRO with $\psi$-divergence.** $\psi$-divergence is one of the most common choices in DRO literature to measure the distance between probability distributions. It encompasses a variety of popular functions such as KL-divergence, $\chi^2$-divergence, and the conditional-value-at-risk (CVaR), etc. Table 1 gives detailed descriptions for these functions.

For CVaR, Namkoong and Duchi [2016] proposed a mirror-descent method which achieves $\mathcal{O}(\sqrt{T})$ regret. Levy et al. [2020] proposed a stochastic gradient-based method with optimal convergence rate in the convex setting. They also discussed an alternative approach based on the dual formulation which they call Dual SGM. In the non-convex setting, Soma and Yoshida [2020] proposed a smoothed approximation of CVaR and obtain an $\mathcal{O}(\epsilon^{-6})$ complexity. We contribute to this line of work by proposing a different divergence with similar behavior as CVaR and an $\mathcal{O}(\epsilon^{-4})$ complexity.

For $\chi^2$ divergence, Hashimoto et al. [2018] considered a constrained formulation of DRO but did not provide theoretical guarantees. Levy et al. [2020] proposed algorithms based on an multi-level Monte-Carlo stochastic gradient estimator, and provide convergence guarantees in the convex setting. In contrast, we consider general smooth non-convex loss function $\ell$ and provide convergence guarantee for $\chi^2$ divergence as a special case of Corollary 3.6.

**Non-smooth non-convex optimization.** Conventional non-convex optimization typically focuses on smooth objective functions. For general smooth non-convex stochastic optimization, it is already known that the best possible gradient complexity for finding an $\epsilon$-approximate stationary point is $\mathcal{O}(\epsilon^{-4})$ [Arjevani et al., 2019], which is achieved by SGD based algorithms [Ghadimi and Lan, 2013]. However, the optimization can be much harder for non-smooth non-convex objective functions, and there are limited results in this setting. Ruszczynski [2020] proposed a stochastic gradient-based method which converges to a stationary point with probability one, under the assumption that the feasible region is bounded. For unconstrained optimization, Zhang et al. [2020c] showed that it is intractable to find an $\epsilon$-stationary point for some Lipschitz and Hadamard semi-differentiable function. When the function is weakly convex, Davis and Drusvyatskiy [2019] showed that the projected SGD converges to the stationary point of a Moreau envelope, and a recent work [Mai and Johansson, 2020] extended this result to SGD with momentum. In this paper, we show that for smooth non-convex loss $\ell$, DRO can be formulated as a non-smooth non-convex optimization problem, but the special property of the DRO objective makes it possible to find an $\epsilon$-stationary point within $\mathcal{O}(\epsilon^{-4})$ complexity.

## 2 Preliminaries

### 2.1 Notations and Assumptions

Throughout this paper we use $\|\cdot\|$ to denote the $\ell_2$-norm in an Euclidean space $\mathbb{R}^d$ and use $\langle\cdot,\cdot\rangle$ to denote the standard inner product. For a real number $t$, denote $(t)_+$ as $\max(t,0)$. For a set $C$, denote $\mathbb{I}_C(\cdot)$ as the indicator function such that $\mathbb{I}_C(x) = 0$ if $x \in C$ and $\mathbb{I}_C(x) = +\infty$ otherwise. We first list some basic definitions in optimization literature, which will be frequently used in this paper.

**Definition 2.1.** *(Lipschitz continuity) A mapping $f : \mathcal{X} \to \mathbb{R}^m$ is G-Lipschitz continuous if for any $x, y \in \mathcal{X}$, $\|f(x) - f(y)\| \leq G\|x - y\|$.*

**Definition 2.2.** *(Smoothness) A function $f : \mathcal{X} \to \mathbb{R}$ is L-smooth if it is differentiable on $\mathcal{X}$ and the gradient $\nabla f$ is L-Lipschitz continuous, i.e. $\|\nabla f(x) - \nabla f(y)\| \leq L\|x - y\|$ for all $x, y \in \mathcal{X}$. We say $f$ is non-smooth if such L does not exist.*

**Definition 2.3.** *(Conjugate function) For a function $\psi : \mathbb{R} \to \mathbb{R}$, the conjugate function $\psi^*$ is defined as $\psi^*(t) := \sup_{s \in \mathbb{R}} (st - \psi(s))$.*

**Assumption 2.4.** *We make the following assumptions throughout the paper:*

- *Given $\xi$, the loss function $\ell(x, \xi)$ is G-Lipschitz continuous and L-smooth with respect to $x$;*

- *$\psi$ is a valid divergence function, i.e. a non-negative convex function satisfying $\psi(1) = 0$ and $\psi(t) = +\infty$ for all $t < 0$. Furthermore the conjugate $\psi^*$ is M-smooth.*

We finally define the notion of $\epsilon$-stationary points for differentiable non-convex functions.

**Definition 2.5.** *($\epsilon$-stationary point) For a differentiable function $f : \mathcal{X} \to \mathbb{R}$, a point $x \in \mathcal{X}$ is said to be first-order $\epsilon$-stationary if $\|\nabla f(x)\| \leq \epsilon$.*

### 2.2 Equivalent formulation of the DRO objective

The aim of this paper is to find an $\epsilon$-stationary point of problem (3). However, the original formulation (3) involves a max operation over distributions which makes optimization challenging. By duality arguments we can show that the DRO objective (3) can be equivalently written as (see detailed derivations in [Levy et al., 2020, Section A.1.2])

$$\Psi(x) = \min_{\eta \in \mathbb{R}} \lambda \mathbb{E}_{\xi \sim P} \psi^* \left( \frac{\ell(x;\xi) - \eta}{\lambda} \right) + \eta. \tag{4}$$

Thus, to minimize $\Psi(x)$ in (4), one can jointly minimize $\mathcal{L}(x,\eta) := \mathbb{E}_{\xi \sim P} \left[ \lambda\psi^* \left( \frac{\ell(x;\xi) - \eta}{\lambda} \right) + \eta \right]$ over $(x, \eta) \in \mathcal{X} \times \mathbb{R} \subset \mathbb{R}^{n+1}$. This can be seen as a standard stochastic optimization problem. The remaining thing is to show one can find an $\epsilon$-stationary point of $\Psi(x)$ by optimizing $\mathcal{L}(x,\eta)$ instead. We first present a lemma that gives connection of the gradient of $\Psi(x)$ to the gradient of $\mathcal{L}(x,\eta)$.

**Lemma 2.6.** *Under the Assumption 2.4, $\Psi(x)$ is differentiable, and $\nabla\Psi(x) = \nabla_x \mathcal{L}(x,\eta)$ for any $\eta \in \arg\min_{\eta'} \mathcal{L}(x,\eta')$.*

Note that the $\eta$ in Lemma 2.6 may not be unique but the values of $\nabla_x \mathcal{L}(x,\eta)$ are all equal. Since $\Psi(x)$ is differentiable, the $\epsilon$-stationary points are well-defined. We now prove that the problem of finding an $\epsilon$-stationary point of $\Psi(x)$ is equivalent to finding an $\epsilon$-stationary point of a rescaled version of $\mathcal{L}(x,\eta)$.

**Theorem 2.7.** *Under the Assumption 2.4, if for some $(x,\eta)$ the following holds: $\|\nabla_x \mathcal{L}(x,\eta)\| + G|\nabla_\eta \mathcal{L}(x,\eta)| \leq \epsilon$, then $x$ is an $\epsilon$-stationary point of $\Psi(x)$. Furthermore, define a rescaled function*

$$\widehat{\mathcal{L}}(x,\eta) = \mathcal{L}(x, G\eta) := \mathbb{E}_{\xi \sim P} \left[ \lambda\psi^* \left( \frac{\ell(x;\xi) - G\eta}{\lambda} \right) + G\eta \right], \tag{5}$$

*then $\|\nabla\widehat{\mathcal{L}}(x,\eta)\| \leq \epsilon/\sqrt{2}$ implies that $x$ is an $\epsilon$-stationary point of $\Psi(x)$.*

The proof of Lemma 2.6 and Theorem 2.7 can be found in Appendix A. From the above theorem it suffices to find an $\epsilon$-stationary point of $\widehat{\mathcal{L}}(x,\eta)$ such that $\|\nabla\widehat{\mathcal{L}}(x,\eta)\| \leq \epsilon$ (ignoring numerical constant $\sqrt{2}$). As a result, we will mainly work with $\widehat{\mathcal{L}}$ in subsequent analysis. The property of the objective function (5) heavily depends on $\psi^*$. We list some popular choices of $\psi$ together with the corresponding $\psi^*$ in Table 1. They serve as motivating examples of our subsequent analysis.

Table 1: Some commonly used divergences and the corresponding conjugates.

| Divergence | $\psi(t)$ | $\psi^*(t)$ |
|---|---|---|
| $\chi^2$ | $\frac{1}{2}(t-1)^2$ | $-1 + \frac{1}{4}(t+2)_+^2$ |
| K-L | $t \log t - t + 1$ | $e^t - 1$ |
| CVaR | $\mathbb{I}_{[0,\alpha^{-1})}, \alpha \in (0,1)$ | $\alpha^{-1}(t)_+$ |
| KL-regularized CVaR | $\mathbb{I}_{[0,\alpha^{-1})} + t \log t - t + 1, \alpha \in (0,1)$ | $\min(e^t, \alpha^{-1}(1 + t + \log \alpha)) - 1$ |
| Cressie-Read | $\frac{t^k - kt + k - 1}{k(k-1)}, k \in \mathbb{R}$ | $\frac{1}{k}\left(((k-1)t+1)_+^{\frac{k}{k-1}} - 1\right)$ |

# 3 Analysis of general non-convex DRO

In this section we will analyze the DRO problem with general smooth non-convex loss functions $\ell$. We first discuss the challenges appearing in our analysis, then show how to leverage the specific structure of the objective function in order to overcome these challenges. Specifically, we show that our proposed algorithm can achieve a non-asymptotic complexity of $\mathcal{O}(\epsilon^{-4})$.

## 3.1 Challenges in non-convex DRO

A standard result in optimization literature states that if the objective function is smooth and the stochastic gradient is unbiased and has bounded variance[3], then standard stochastic gradient descent (SGD) algorithms can provably find an $\epsilon$-first-order stationary point under $\mathcal{O}(\epsilon^{-4})$ gradient complexity [Ghadimi and Lan, 2013]. Here the smoothness and bounded variance property are crucial for the convergence of SGD [Zhang et al., 2019]. However, we find that *both* assumptions are violated in non-convex DRO, even if the *inner* loss $\ell(x,\xi)$ is smooth and the stochastic noise is bounded for both $\ell(x,\cdot)$ and $\nabla_x \ell(x,\cdot)$. We present a counter example to illustrate this point, in which we can gain some insight about the structure of the DRO objective.

**Example 3.1.** Consider the loss $\ell(x;\xi) = x^2 \left(1 + \frac{\xi}{x^2+1}\right)^2$ which is a quadratic-like function with noise $\xi$, where $\xi$ is a Rademacher variable drawn from $\{-1,+1\}$ with equal probabilities. Then a straightforward calculation shows that the loss $\ell$ has the following properties:

- (Smoothness) For any $\xi \in \{-1,+1\}$, $\ell(x,\xi)$ is $L$-smooth with respect to $x$ for $L = 8$;

- (Bounded variance) For any $x \in \mathbb{R}$, $\mathbb{E}_\xi\left[\left(\ell(x,\xi) - x^2\right)^2\right] = \frac{4x^4}{(x^2+1)^2} + \frac{x^4}{(x^2+1)^4} \leq 4$. It then follows that $\text{Var}_\xi[\ell(x,\xi)] \leq 4$;

- (Bounded variance for gradient) Similarly we can check that the gradient of $\ell$ also has bounded variance. Moreover, the variance tends to zero when $x \to \infty$.

Now consider the DRO where $\psi$ is chosen as the commonly used $\chi^2$-divergence. Fix $\lambda = 1$ and $\eta = 0$. Based on the expression of $\psi^*(t)$ in Table 1, the DRO objective function (5) thus takes the form $\widehat{\mathcal{L}}(x,0;\xi) = \frac{1}{4}\left[x^2\left(1 + \frac{\xi}{x^2+1}\right)^2 + 2\right]^2 - 1$, which is a quartic-like function. It follows that

- $\widehat{\mathcal{L}}(x,0;\xi) = \Theta(x^4)$ for large $x$ and therefore $\widehat{\mathcal{L}}(x,0;\xi)$ is not globally smooth;

- $\nabla_x\widehat{\mathcal{L}}(x,0;\xi) = x^3 + 2x\xi + 2x + \mathcal{O}(1)$ for large $x$ and the stochastic gradient variance $\text{Var}[\nabla_x\widehat{\mathcal{L}}(x,0;\xi)] = \Theta(x^2)$ which is unbounded globally.

As we can see from the above example, both the local smoothness and the gradient variance of $\widehat{\mathcal{L}}$ strongly rely on the scale of $x$. Indeed, in general non-convex DRO both the two quantities have a positive correlation with the magnitude of $\ell$. As shown in Appendix B, if we make the additional assumption that $\ell$ is bounded by a small constant, then the smoothness and gradient noise can be controlled in a straightforward way, and we show that a projected stochastic gradient method can be applied in this setting. However, such bounded loss assumption is quite restrictive and not satisfactory.

---

[3]$\mathbb{E}_{\xi \sim P}\|\nabla_x \ell(x,\xi) - \nabla_x \ell(x)\|^2 \leq \sigma^2$ for some $\sigma$ and all $x \in \mathcal{X}$ where $\ell(x) = \mathbb{E}_{\xi \sim P}\ell(x,\xi)$.

## 3.2 Main results

In this section, we present the main theoretical result of this paper. All proofs can be founded in Appendix C. We make the following assumption on the noise of the stochastic loss:

**Assumption 3.2.** *We assume that for all $x \in \mathcal{X}$, the stochastic loss has bounded variance, i.e. $\mathbb{E}_{\xi \sim P} \left( \ell(x, \xi) - \ell(x) \right)^2 \leq \sigma^2$ where $\ell(x) = \mathbb{E}_{\xi \sim P} \ell(x, \xi)$.*

We now provide formal statements of the key properties mentioned above, which show that both the gradient variance and the local smoothness can be controlled in terms of the gradient norm.

**Lemma 3.3.** *Under Assumptions 2.4 and 3.2, the gradient estimators of (5) satisfies the following property:*

$$\mathbb{E}_\xi \|\nabla \widehat{\mathcal{L}}(x, \eta, \xi) - \nabla \widehat{\mathcal{L}}(x, \eta)\|^2 \leq 11 G^2 M^2 \lambda^{-2} \sigma^2 + 8(G^2 + \|\nabla \widehat{\mathcal{L}}(x, \eta)\|^2) \quad (6)$$

**Lemma 3.4.** *Under Assumption 2.4, for any pair of parameters $(x, \eta)$ and $(x', \eta')$, we have the following property for the gradient of $\widehat{\mathcal{L}}$:*

$$\|\nabla \widehat{\mathcal{L}}(x, \eta) - \nabla \widehat{\mathcal{L}}(x', \eta')\| \leq \left( K + \tfrac{L}{G} \|\nabla \widehat{\mathcal{L}}(x, \eta)\| \right) \|(x - x', \eta - \eta')\| \quad (7)$$

*where $K = L + 2G^2 \lambda^{-1} M$.*

Note that (7) reduces to the standard notion of smoothness if the term $\frac{L}{G} \|\nabla \widehat{\mathcal{L}}(x, \eta)\|$ is absent. Thus the inequality (7) can be seen as a generalized smoothness condition. Zhang et al. [2020b] for the first time proposed such generalized smoothness for twice-differentiable functions in a different form, and Zhang et al. [2020a] further gave a comprehensive analysis of algorithms for optimizing generalized smooth functions. However, all these works make strong assumptions on the gradient noise and can not be applied in our setting.

Instead, we propose to use the *mini-batch normalized SGD with momentum* algorithm for non-convex DRO, shown in Algorithm 1. The algorithm has been theoretically analysed in [Cutkosky and Mehta, 2020] for optimizing standard smooth non-convex functions. Compared with Cutkosky and Mehta [2020], we use mini-batches in each iteration in order to ensure convergence in our setting.

---

**Algorithm 1:** Mini-batch Normalized SGD with Momentum

**Input :** The objective function $F(w)$, distribution $P$, initial point $w_0$, initial momentum $m_0$, learning rate $\gamma$, momentum factor $\beta$, batch size $S$ and total number of iterations $T$

1   **for** $t \leftarrow 1$ **to** $T$ **do**

2     $\hat{\nabla} F(w_{t-1}) \leftarrow \frac{1}{S} \sum_{i=1}^{S} \nabla F(w_{t-1}, \xi_{t-1}^{(i)})$ where $\{\xi_{t-1}^{(i)}\}_{i=1}^{S}$ are i.i.d. samples drawn from $P$

3     $m_t \leftarrow \beta m_{t-1} + (1 - \beta) \hat{\nabla} F(w_{t-1})$

4     $w_t \leftarrow w_{t-1} - \gamma \dfrac{m_t}{\|m_t\|}$

---

The following main theorem establishes convergence guarantee of Algorithm 1. We further provide a sketch of proof in Section 3.3, where we can gain insights on how normalization and momentum techniques help tackle the difficulties shown in Lemmas 3.3 and 3.4.

**Theorem 3.5.** *Suppose that $F$ satisfies the following conditions:*

- *(Generalized smoothness) $\|\nabla F(w_1) - \nabla F(w_2)\| \leq (K_0 + K_1 \|\nabla F(w_1)\|) \|w_1 - w_2\|$ holds for any $w_1, w_2$;*

- *(Gradient variance) The stochastic gradient $\nabla F(w, \xi)$ is unbiased ($\nabla F(w) = \mathbb{E}_\xi \nabla F(w, \xi)$) and satisfies $\mathbb{E}_\xi \|\nabla F(w, \xi) - \nabla F(w)\|^2 \leq \Gamma^2 \|\nabla F(w)\|^2 + \Lambda^2$ for some $\Gamma$ and $\Lambda$.*

*Let $\{w_t\}$ be the sequence produced by Algorithm 1. Then with a mini-batch size $S = \Theta(\Gamma^2)$ and a suitable choice of parameters $\gamma$ and $\beta$, for any small $\epsilon = \mathcal{O}(\min(K_0/K_1, \Lambda/\Gamma))$, we need at most $\mathcal{O}\left( \Delta K_0 \Lambda^2 \epsilon^{-4} \right)$ gradient complexity to guarantee that we find an $\epsilon$-stationary point in expectation, i.e. $\frac{1}{T} \sum_{t=0}^{T-1} \mathbb{E} \|\nabla F(w_t)\| \leq \epsilon$ where $\Delta = F(w_0) - \inf_{w \in \mathbb{R}^d} F(w)$.*

Substituting Lemmas 3.3 and 3.4 into Theorem 3.5 immediately yields the final result:

**Corollary 3.6.** *Suppose the DRO problem* (3) *satisfies Assumptions 2.4 and 3.2. Using Algorithm 1 with a constant batch size, the gradient complexity for finding an $\epsilon$-stationary point of $\Psi(x)$ is*

$$\mathcal{O}\left(G^2\left(M^2\sigma^2\lambda^{-2}+1\right)\left(\lambda^{-1}MG^2+L\right)\Delta\epsilon^{-4}\right).$$

Corollary 3.6 shows that Algorithm 1 finds an $\epsilon$-stationary point with complexity $\mathcal{O}(\epsilon^{-4})$, which is the same as standard smooth non-convex optimization. Also note that the bound in Theorem 3.5 does not depend on $K_1$ and $\Gamma$ as long as $\epsilon$ is sufficiently small. In other words, Algorithm 1 is well-adapted to the non-smoothness and unbounded noise in our setting. We also point out that although the batch size is chosen propositional to $\Gamma^2$, the required number of iterations $T$ is inversely propositional to $\Gamma^2$, therefore the total number of stochastic gradient computations remains the same.

Finally, note that Theorem 3.5 is stated in a general form and is not limited to DRO setting. It greatly extends the results in Zhang et al. [2020a,b] by relaxing their noise assumptions, and demonstrates the effectiveness of combining adaptive gradients with momentum for optimizing ill-conditioned objective functions. More importantly, our algorithm is to some extent similar to currently widely used optimizers in practice, e.g. Adam. We believe our result can shed light on why these optimizers often show superior performance in real applications.

### 3.3 Proof sketch of Theorem 3.5

Below we present our proof sketch, in which the motivation of using Algorithm 1 will be clear. Similar to standard analysis in non-convex optimization, we first derive a descent inequality for functions satisfying the generalized smoothness:

**Lemma 3.7.** *(Descent inequality) Let $F(x)$ be a function satisfying the generalized smoothness condition in Theorem 3.5. Then for any point $x$ and direction $z$ the following holds:*

$$F(x-z) \leq F(x) - \langle \nabla F(x), z \rangle + \frac{1}{2}(K_0 + K_1\|\nabla F(x)\|)\|z\|^2. \tag{8}$$

The above lemma suggests that the algorithm should take a small step size when $\|\nabla F(x)\|$ is large in order to decrease $F$. This is the main motivation of considering a normalized update. Indeed, after some careful calculation we can prove the following result:

**Lemma 3.8.** *Consider the algorithm that starts at $w_0$ and makes updates $w_{t+1} = w_t - \gamma\frac{m_{t+1}}{\|m_{t+1}\|}$ where $\{m_t\}$ is an arbitrary sequence of points. Define $\delta_t := m_{t+1} - \nabla F(w_t)$ be the estimation error. If $\gamma = O(1/K_1)$, then*

$$F(w_t) - F(w_{t+1}) \geq \left(\gamma - \frac{1}{2}K_1\gamma^2\right)\|\nabla F(w_t)\| - \frac{1}{2}K_0\gamma^2 - 2\gamma\|\delta_t\| \tag{9}$$

which is $\gamma\|\nabla F(w_t)\| - 2\gamma\|\delta_t\| - \mathcal{O}(\gamma^2)$ for small $\gamma$. Therefore the objective function $F(w)$ decreases if $\|\delta_t\| < 1/2 \cdot \|\nabla F(w_t)\|$, i.e. a small estimation error. However, $\delta_t$ is related to the stochastic gradient noise which can be very large due to Lemma 3.3. This motivates us to the use the momentum technique for the choice of $\{m_t\}$ to reduce the noise. Formally, let $\beta$ be the momentum factor and define $\hat{\delta}_t = \hat{\nabla}F(w_t) - \nabla F(w_t)$, then using the recursive equation of momentum $m_t$ in Algorithm 1 we can show that

$$\delta_t = \beta\sum_{\tau=0}^{t-1}\beta^\tau(\nabla F(w_{t-\tau-1}) - \nabla F(w_{t-\tau})) + (1-\beta)\sum_{\tau=0}^{t-1}\beta^\tau\hat{\delta}_{t-\tau} + (1-\beta)\beta^t\hat{\delta}_0. \tag{10}$$

The first term of the right hand side in (10) can be bounded using the generalized smoothness condition, and the core procedure is to bound the second term using a careful analysis of conditional expectation and the independence of noises $\{\hat{\delta}_t\}$ (see Lemma C.9 in Appendix). Finally, the use of mini-batches of size $\Theta(\Gamma^2)$, a carefully chosen $\beta$ and a small enough $\gamma$ ensure that $\sum_{t=0}^{T-1}\|\delta_t\| < c\sum_{t=0}^{T-1}(\mathbb{E}\|\nabla F(w_t)\| + \mathcal{O}(\epsilon))$ where $c < 1/2$. This guarantees that the right hand side of (9) is overall positive, and by taking summation over $t$ in (9) we have that

$$F(w_0) - F(w_T) \geq (1-2c)\gamma\sum_{t=0}^{T-1}\|\nabla F(w_t)\| - \mathcal{O}(\gamma^2 T - \gamma T\epsilon).$$

namely,
$$\frac{1}{T}\sum_{t=0}^{T-1}\|\nabla F(w_t)\| \leq \mathcal{O}\left(\frac{\Delta}{\gamma T} + \gamma + \epsilon\right).$$

Finally, for a suitable choice of $\gamma$ we can obtain the minimum gradient complexity bound on $T$.

## 3.4 Dealing with the CVaR case

Previous analysis applies to any divergence function $\psi$ as long as $\psi^*$ is smooth. This includes some popular choices such as the $\chi^2$-divergence, but not the CVaR. In the case of CVaR, $\psi^*$ is not differentiable as shown in Table 1, which is undesirable from an optimization viewpoint. In this section we introduce a smoothed version of CVaR. The conjugate function of the smoothed CVaR is also smooth, so that the results in Section 3.2 can be directly applied in this setting.

For standard CVaR at level $\alpha$, $\psi_\alpha(t)$ takes zero when $t \in [0, 1/\alpha)$ and takes infinity otherwise. Instead, we consider the following smoothed version of CVaR:

$$\psi_\alpha^{\text{smo}}(t) = \begin{cases} t \log t + \frac{1-\alpha t}{\alpha} \log \frac{1-\alpha t}{1-\alpha} & t \in [0, 1/\alpha) \\ +\infty & \text{otherwise} \end{cases} \tag{11}$$

It is easy to see that $\psi_\alpha^{\text{smo}}$ is a valid divergence. The corresponding conjugate function is

$$\psi_\alpha^{\text{smo},*}(t) = \frac{1}{\alpha} \log(1 - \alpha + \alpha \exp(t)). \tag{12}$$

The following propositions demonstrate that $\psi_\alpha^{\text{smo}}$ is indeed a smoothed approximation of CVaR.

**Proposition 3.9.** *Fix $0 < \alpha < 1$. When $\lambda \to 0^+$, the solution of the DRO problem (5) for smoothed CVaR tends to the solution for the standard CVaR. Note that the solution of the standard CVaR does not depend on $\lambda$.*

**Proposition 3.10.** $\psi_\alpha^{smo,*}(t)$ *is $\frac{1}{\alpha}$-Lipschitz and $\frac{1}{4\alpha}$-smooth.*

Based on Proposition 3.10, we can then use Corollary 3.6 to obtain the gradient complexity (taking $M = 1/4\alpha$).

Note that $\psi_\alpha^{\text{smo},*}(t)$ is not only smooth but also Lipschitz. In this setting, we can in fact obtain a stronger result than the general one provided in Corollary 3.6. Specifically, the gradient noise and smoothness of the objective function $\widehat{\mathcal{L}}(x, \eta, \xi)$ can be bounded, as shown in the following lemma:

**Lemma 3.11.** *Suppose Assumption 2.4 holds. For smoothed CVaR, the DRO objective (5) satisfies*

$$\mathbb{E}\|\nabla\widehat{\mathcal{L}}(x, \eta, \xi)\|^2 \leq 2\alpha^{-2}G^2. \tag{13}$$

*Moreover, $\widehat{\mathcal{L}}(x, \eta)$ is $K$-smooth with $K = \frac{L}{\alpha} + \frac{G^2}{2\lambda\alpha}$.*

Equipped with the above lemma, we can obtain the following guarantee for smoothed CVaR, which shows that *vanilla SGD* suffices for convergence.

**Theorem 3.12.** *Suppose that $\psi = \psi_\alpha^{smo}$ and Assumption 2.4 holds. If we run SGD with properly selected hyper-parameters on the loss $\widehat{\mathcal{L}}(x, \eta)$, then the gradient complexity of finding an $\epsilon$-stationary point of $\Psi(x)$ is $\mathcal{O}\left(\alpha^{-3}\lambda^{-1}G^2(G^2 + \lambda L)\Delta\epsilon^{-4}\right)$, where $\Delta = \mathcal{L}(x_0, \eta_0) - \inf_x \Psi(x)$.*

The above theorem shows a similar convergence rate compared with Corollary 3.6 in terms of $\epsilon$ and $G$, and the dependency on $\lambda$ is even better. Therefore the Lipschitz property of $\psi^*$ is very useful, in that it is now possible to use a simpler algorithm while achieving a similar (or even better) bound.

## 4 Experiments

We perform two sets of experiments to verify our theoretical results. In the first set of experiments, we consider the setting in Section 3.2, where the loss $\ell(x; \xi)$ is highly non-convex and unbounded, and $\psi$ is chosen to be the commonly used $\chi^2$-divergence such that its conjugate is smooth. We will show that (i) the vanilla SGD algorithm cannot optimize this loss efficiently due to the non-smoothness of the DRO objective; (ii) by simply adopting the normalized momentum algorithm, the optimization process can be greatly accelerated. In the second set of experiments, we deal with the CVaR setting in Section 3.4. We will show that by employing the smooth approximation of CVaR defined in (11) and (12), the optimization can be greatly accelerated.

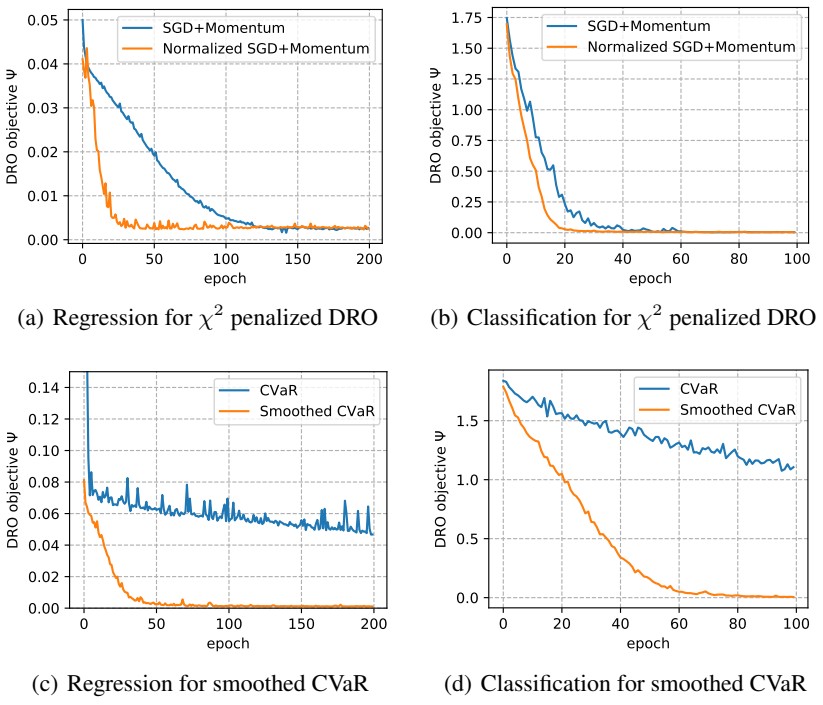

(a) Regression for $\chi^2$ penalized DRO

(b) Classification for $\chi^2$ penalized DRO

(c) Regression for smoothed CVaR

(d) Classification for smoothed CVaR

Figure 1: Training curve of $\chi^2$ penalized DRO and smoothed CVaR in regression and classification task.

## 4.1 Experimental settings

**Tasks.** We consider two tasks: the classification task and the regression task. While classification is more common in machine learning, here we may also highlight the regression task, since recent studies show that DRO may be more suitable for non-classification problems in which the metric of interest is continuous as opposed to the 0-1 loss [Hu et al., 2018, Levy et al., 2020].

**Datasets.** We choose the AFAD-Full dataset for regression and CIFAR-10 dataset for classification. AFAD-Full [Niu et al., 2016] is a regression task to predict the age of human from the facial information, which contains more than 160K facial images and the corresponding age labels ranging from 15 to 75. Note that AFAD-Full is an imbalanced dataset where the ages of two thirds of the whole dataset are between 18 and 30. Following the experimental setting in [Chang et al., 2011, Chen et al., 2013, Niu et al., 2016], we split the whole dataset into a training set comprised of 80% data and a test set comprised of the remaining 20% data. CIFAR-10 dataset is a classification task consisting of 10 classes with 5000 images for each class. To demonstrate the effectiveness of our method in DRO setting, we adopt the setting in Chou et al. [2020] to construct an imbalanced CIFAR-10 by randomly sampling each category at different ratio. See Appendix for more details.

**Model.** For all experiments in this paper, we use the standard ResNet-18 model in [He et al., 2016]. The output has 10 logits for CIFAR-10 classification task, and has a single logit for regression.

**Training details.** We choose the penalty coefficient $\lambda = 0.1$ and the CVaR coefficient $\alpha = 0.02$ in all experiments. For each algorithm, we tune the learning rate hyper-parameter from a grid search and pick the one that achieves the fastest optimization speed. The momentum factor is taken to 0.9 in all experiments, and the mini-batch size is chosen to be 128. We train the model for 100 epochs on CIFAR-10 dataset and 200 epochs on AFAD-Full dataset. Other training details can be found in Appendix E.

## 4.2 Experimental results

Results are demonstrated in Figure 1. For each figure, we plot the value of the DRO objective $\Psi(x)$ through the training process. Here we calculate $\Psi(x) = \min_\eta \mathcal{L}(x, \eta)$ at each epoch based on a convex optimization on $\eta$ until convergence (rather than using $\mathcal{L}(x, \eta)$ with the current parameter $\eta$ directly).

**Experimental result for $\chi^2$ penalized DRO.** Figure 1(a) and Figure 1(b) plot the training curve of the DRO objective using different algorithms. It can be seen that in both regression and classification, vanilla SGD converges slowly, and using normalized momentum algorithm significantly improves the convergence speed. For example, in regression task SGD does not converge after 100 epochs while normalized momentum algorithm converges just after 25 epochs. These results highly consist with our theoretical findings, which shows that due to the non-smoothness of the DRO loss, vanilla SGD may not be able to optimize the loss well; In contrast, normalized momentum utilizes the relationship between local smoothness and gradient magnitude, and achieves better performance.

**Experimental result for smoothed CVaR.** Figure 1(c) and Figure 1(d) plot the training curves for different training losses: CVaR and smoothed CVaR. Note that the evaluation metrics ($y$-axis) in these figures are all chosen to be CVaR, even when the training objective is smoothed CVaR. In this way we can make a fair comparison of optimization speed based on these training curves. Firstly, it can be seen that the optimization of CVaR is very hard due to the non-smoothness, and the training curves have lots of spikes. In contrast, the optimization of smoothed CVaR is much easier for both tasks, and the final loss is significantly lower. Such experimental results show the benefit of our proposed smoothed CVaR for optimization.

**Test performance**. We also measure the test performance of trained models to see whether a better optimizer can also improve test accuracy. Due to space limitation, in the main text we provide results of $\chi^2$ penalized DRO problem for classification using unbalanced CIFAR-10 dataset, which is listed in Table 2. Other results can be found in Appendix E. It can be seen that the model trained using normalized SGD with momentum achieves higher test accuracy on all class, and especially, the worst-performing class. Since the experiments in this paper is mainly designed to compare algorithms rather than to achieve best performance, better performance is likely to be reached if adjusting the hyper-parameters (e.g. $\lambda$, the number of epochs, and the learning rate schedule).

Table 2: Test performance of the $\chi^2$ penalized DRO problem for unbalanced CIFAR-10 classification. Each column corresponds to the performance of a particular class. The bolded column indicates the worst-performing class.

| Class | 1 | 2 | 3 | 4 | 5 | **6** | 7 | 8 | 9 | 10 |
|---|---|---|---|---|---|---|---|---|---|---|
| Number of training samples | 4020 | 2715 | 4985 | 2965 | 1950 | **1425** | 4795 | 4030 | 4835 | 3300 |
| Test acc (SGD+Momentum) | 76.7 | 80.1 | 70.2 | 55.0 | 54.6 | **44.8** | 84.9 | 77.7 | 85.5 | 76.8 |
| Test acc (Normalized SGD+Mom.) | 78.8 | 81.2 | 71.7 | 57.3 | 56.2 | **49.8** | 87.2 | 83.5 | 90.4 | 78.4 |

## 5 Discussion

**Conclusion.** In this paper we provide non-asymptotic analysis of first-order algorithms for the DRO problem with unbounded and non-convex loss. Specifically, we write the original DRO problem as a non-smooth non-convex optimization problem, and we propose an efficient normalization-based algorithm to solve it. The general result of Theorem 3.5 might be of independent value and is not limited to DRO setting. We hope that this work can also bring inspiration to the study of other non-smooth non-convex optimization problems.

**Limitations.** Despite the theoretical grounds and promising experimental justifications, there are some interesting questions that remain unexplored. Firstly, it may be possible to obtain better complexities on problem-dependent parameters, e.g. $G$ and $\lambda$. Secondly, while this paper mainly considers smooth $\psi^*$, in some cases $\psi^*$ may be non-smooth (e.g. for KL-divergence) or even not continuous. In future we hope to discover approaches that can deal with more general classes of $\psi$-divergence. Finally, we are looking forward to seeing more applications of DRO in real-world problems.

## Acknowledgement

This work was supported by Key-Area Research and Development Program of Guangdong Province (No. 2019B121204008), National Key R&D Program of China (2018YFB1402600), BJNSF (L172037) and Beijing Academy of Artificial Intelligence. Project 2020BD006 supported by PKU-Baidu Fund. Jikai Jin is partially supported by the elite undergraduate training program of School of Mathematical Sciences in Peking University.

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
