# A  Equivalent formulation of the DRO objective

## A.1  Generalized gradient

As can be seen, the problem (4) is the pointwise minima over $\eta$ for a family of smooth functions $\mathcal{L}(x, \eta)$. However, there exists a known result showing that the pointwise minima of a family of smooth functions may not be differentiable in general, so the gradient may not exist[4].

We first assume $\Psi(x)$ is *non-smooth* and non-convex. To measure the convergence of non-smooth non-convex optimization, we define the notion called the *generalized gradient* [Clarke, 1990, Chapter 2].

**Definition A.1.** *(Local Lipschitzness) A function $f : \mathcal{X} \to \mathbb{R}^m$ is locally Lipschitz continuous near point $x \in \text{int}(\mathcal{X})$ if there exists $G, \epsilon > 0$ such that for any $y, z \in \mathcal{B}_\epsilon(x)$, $|f(y) - f(z)| \leq G\|y - z\|$. Here $\mathcal{B}_\epsilon(x)$ denotes the set of points in the open ball of radius $\epsilon$ around $x$.*

**Definition A.2.** *(Generalized gradient) Suppose that $f : \mathcal{X} \to \mathbb{R}$ is locally Lipschitz-continuous at $x$, where $\mathcal{X} \subset \mathbb{R}^n$. Its generalized directional derivative in direction $v$ is defined as*

$$f^\circ(x; v) = \limsup_{\substack{y \to x \\ t \to 0}} \frac{f(y + tv) - f(y)}{t}$$

*and the generalized gradient at $x$ is the set*

$$\partial f(x) = \{\zeta \in \mathbb{R}^n : f^\circ(x; v) \geq \langle \zeta, v \rangle \, \forall v \in \mathbb{R}^n\}.$$

Interested readers may refer to the book [Clarke, 1990] for an in-depth exploration of this concept. Importantly, $\partial f(x)$ is a non-empty closed convex set; $\partial f(x)$ degenerates to a single point $\{\nabla f(x)\}$ if $f$ is smooth, and $\partial f(x)$ is equivalent to the sub-gradient if $f$ is convex. If $x$ is a local minima (or maxima) for $f(x)$, then $0 \in \partial f(x)$. The following proposition gives the relationship between generalized gradient and (conventional) gradient.

**Proposition A.3.** *([Clarke, 1990, Section 2.2]) If function $f$ is differentiable at $x$, then $f$ is local Lipschitz near $x$ and $\partial f(x) = \{\nabla f(x)\}$. Conversely, if $f$ is local Lipschitz near $x$ and $\partial f(x)$ reduces to a singleton point $\{g\}$, then $f$ is differentiable at $x$ and $\nabla f(x) = g$.*

## A.2  Proof of Lemma 2.6

We first present a basic lemma which provides a rule to calculate generalized gradients of the pointwise maxima of a function family.

**Lemma A.4** ([Clarke, 1981]). *Suppose that $\mathcal{T} \subset \mathbb{R}^m$ is compact and $\mathcal{X} \subset \mathbb{R}^n$ is open. Let $f : \mathcal{X} \times \mathcal{T} \to \mathbb{R}$ be a $K$-Lipschitz continuous function in $x \in \mathcal{X}$ for some $K$ and is continuous in $t \in \mathcal{T}$. Define the point-wise minima function $F(x) = \min_{t \in \mathcal{T}} f(x, t)$, then we have*

$$\partial F(x) \subset \text{Conv} \bigcup_{t \in T(x)} \partial_x f(x, t) \tag{14}$$

*where $T(x) = \{t \in T : F(x) = f(x, t)\}$, $\partial_x f(x, t)$ is the partial generalized gradient and $\text{Conv}$ denotes a convex hull of a point set.*

Recall that in our setting $\Psi(x) = \min_{\eta \in \mathbb{R}} \mathcal{L}(x, \eta)$. Since $\psi^*$ and $\ell$ are differentiable, $\mathcal{L}$ is differentiable in both $\eta$ and $x$. To make use of Lemma A.4, we have to constrain $\eta$ in a compact set $\mathcal{T}$. This is possible if we constrain $x$ in a compact set $\mathcal{B}_r(x_0)$, an open ball of radius $r$ centered at $x_0$.

**Lemma A.5.** *Assume Assumption 2.4 holds. Fix a point $x_0 \in \mathcal{X}$. Denote $\eta_0 \in \text{argmin}_\eta \mathcal{L}(x_0, \eta)$ be an arbitrary minima. Then for any point $x \in \mathcal{B}_r(x_0)$ near $x_0$, there exists $\eta_x \in \text{argmin}_\eta \mathcal{L}(x, \eta)$, such that $|\eta_0 - \eta_x| \leq Gr$.*

---

[4]For example, consider function $f(x, \eta) = \frac{1}{(\eta^2+1)} \log(1 + \exp((\eta^2 + 1)x))$ that is jointly smooth in $(x, \eta)$. However, the pointwise minima $\min_{\eta \in \mathbb{R}} f(x, \eta) = \max(x, 0)$ which is non-differentiable at $x = 0$.

**Proof:** Using the condition that $\psi^*$ is convex and differentiable, we have $\eta_x \in \operatorname{argmin}_\eta \mathcal{L}(x, \eta)$ if and only if $\nabla_\eta \mathcal{L}(x, \eta) = 0$. Namely,

$$\nabla_\eta \mathcal{L}(x, \eta) = 1 - \mathbb{E}_\xi \left[ (\psi^*)' \left( \frac{\ell(x; \xi) - \eta_x}{\lambda} \right) \right] = 0 \quad \text{iff} \quad \eta_x \in \operatorname{argmin}_\eta \mathcal{L}(x, \eta). \tag{15}$$

For any point $x \in \mathcal{B}_r(x_0)$, $|\ell(x; \xi) - \ell(x_0; \xi)| \leq Gr$ holds for any $\xi$ due to the Lipschitz property of $\ell(\cdot; \xi)$. Considering that $(\psi^*)'$ is monotonically increasing (due to the convexity of $\psi^*$), we have

$$1 - \mathbb{E}_\xi \left[ (\psi^*)' \left( \frac{\ell(x; \xi) - (\eta_0 + Gr)}{\lambda} \right) \right] \geq 0 \tag{16}$$

$$1 - \mathbb{E}_\xi \left[ (\psi^*)' \left( \frac{\ell(x; \xi) - (\eta_0 - Gr)}{\lambda} \right) \right] \leq 0 \tag{17}$$

Since $\nabla_\eta \mathcal{L}(x, \eta)$ is continuous in $\eta$, there must exists an $\eta_x \in [\eta_0 - Gr, \eta_0 + Gr]$, such that

$$1 - \mathbb{E}_\xi \left[ (\psi^*)' \left( \frac{\ell(x; \xi) - \eta_x}{\lambda} \right) \right] = 0 \tag{18}$$

Therefore $\eta_x \in \operatorname{argmin}_\eta \mathcal{L}(x, \eta)$. $\qquad \square$

For any point $x_0$, we can use Lemma A.4 by substituting $\mathcal{X} = \mathcal{B}_r(x_0)$ and $\mathcal{T} = [\eta_0 - Gr, \eta_0 + Gr]$. The procedure is as follows:

- $\Psi(x) = \min_{\eta \in \mathbb{R}} \mathcal{L}(x, \eta) = \min_{\eta \in [\eta_0 - Gr, \eta_0 + Gr]} \mathcal{L}(x, \eta)$ holds for all $x \in \mathcal{B}_r(x_0)$;
- Applying Lemma A.4 we obtain

$$\partial \Psi(x) \subset \operatorname{Conv}\{\nabla_x \mathcal{L}(x, \eta) : \eta \in [\eta_0 - Gr, \eta_0 + Gr] \cap \operatorname{argmin}_\eta \mathcal{L}(x, \eta)\}$$
$$\subset \operatorname{Conv}\{\nabla_x \mathcal{L}(x, \eta) : \eta \in \operatorname{argmin}_\eta \mathcal{L}(x, \eta)\}$$

We finally prove below (Lemma A.6) that $\{\nabla_x \mathcal{L}(x, \eta) : \eta \in \operatorname{argmin}_\eta \mathcal{L}(x, \eta)\}$ is a singleton set. Then Proposition A.3 indicates that $\Psi(x)$ is differentiable, and the generalized gradient reduces to gradient such that $\nabla \Psi(x) = \nabla_x \mathcal{L}(x, \eta)$ for any $\eta \in \operatorname{argmin}_\eta \mathcal{L}(x, \eta)$. Thus we complete the proof of Lemma 2.6.

**Lemma A.6.** *Assume Assumption 2.4 holds. For any $\eta_1, \eta_2 \in \operatorname{argmin}_\eta \mathcal{L}(x, \eta)$, we have $\nabla_x \mathcal{L}(x, \eta_1) = \nabla_x \mathcal{L}(x, \eta_2)$.*

**Proof:** Denote $X(x, \eta), Y(x)$ be two random functions defined by

$$X(x, \eta) = (\psi^*)' \left( \frac{\ell(x; \xi) - \eta}{\lambda} \right) \quad Y(x) = \nabla_x \ell(x; \xi)$$

which depend on the random variable $\xi$. Rewrite the gradient of $\mathcal{L}(x, \eta)$ as follows:

$$\nabla_x \mathcal{L}(x, \eta) = \mathbb{E}[X(x, \eta) Y(x)] \tag{19}$$
$$\nabla_\eta \mathcal{L}(x, \eta) = 1 - \mathbb{E}[X(x, \eta)] \tag{20}$$

Note that $(\psi^*)'$ is monotonically increasing (due to the convexity of $\psi^*$), thus $X(x, \eta)$ is monotonically decreasing in $\eta$. It follows that

$$\nabla_\eta \mathcal{L}(x, \eta_1) = \nabla_\eta \mathcal{L}(x, \eta_2) \quad \text{iff} \quad \mathbb{E}[X(x, \eta_1)] = \mathbb{E}[X(x, \eta_2)] \quad \text{iff} \quad X(x, \eta_1) = X(x, \eta_2) \quad \text{a.s.}$$

Therefore $\mathbb{E}[X(x, \eta_1) Y(x)] = \mathbb{E}[X(x, \eta_2) Y(x)]$, namely $\nabla_x \mathcal{L}(x, \eta_1) = \nabla_x \mathcal{L}(x, \eta_2)$. $\qquad \square$

### A.3 Proof of Theorem 2.7

Now, suppose that we have obtained a pair $(x, \eta)$ s.t. $\|\nabla_x \mathcal{L}(x, \eta)\| + G |\nabla_\eta \mathcal{L}(x, \eta)| \leq \epsilon$. Let $x$ be fixed and $\eta^* \in \arg\min_\eta \mathcal{L}(x, \eta)$. Then we have

$$\|\nabla_x \mathcal{L}(x, \eta) - \nabla_x \mathcal{L}(x, \eta^*)\|$$
$$= \left\| \mathbb{E}_\xi \left[ \left( (\psi^*)' \left( \frac{\ell(x; \xi) - \eta}{\lambda} \right) - (\psi^*)' \left( \frac{\ell(x; \xi) - \eta^*}{\lambda} \right) \right) \nabla \ell(x; \xi) \right] \right\|$$
$$\leq G \cdot \mathbb{E}_\xi \left| (\psi^*)' \left( \frac{\ell(x; \xi) - \eta}{\lambda} \right) - (\psi^*)' \left( \frac{\ell(x; \xi) - \eta^*}{\lambda} \right) \right|$$
$$= G \cdot \left| \mathbb{E}_\xi \left[ (\psi^*)' \left( \frac{\ell(x; \xi) - \eta}{\lambda} \right) - (\psi^*)' \left( \frac{\ell(x; \xi) - \eta^*}{\lambda} \right) \right] \right|$$
$$= G |\nabla_\eta \mathcal{L}(x, \eta) - \nabla_\eta \mathcal{L}(x, \eta^*)| = G |\nabla_\eta \mathcal{L}(x, \eta)|$$

where we use the fact that $(\psi^*)'$ is monotone increasing (due to the comvexity of $\psi^*$. Hence, using Lemma 2.6 we obtain

$$\|\nabla\Psi(x)\| = \|\nabla_x\mathcal{L}(x,\eta^*)\| \leq \|\nabla_x\mathcal{L}(x,\eta)\| + G\,|\nabla_\eta\mathcal{L}(x,\eta)| \leq \epsilon$$

Now suppose that $\|\nabla\widehat{\mathcal{L}}(x,\eta)\| \leq \epsilon/\sqrt{2}$. Then

$$\|\nabla\widehat{\mathcal{L}}(x,\eta)\|^2 = \|\nabla_x\mathcal{L}(x,G\eta)\|^2 + G^2|\nabla_\eta\mathcal{L}(x,G\eta)|^2 \leq \epsilon^2/2$$

Using $(a+b)^2 \leq 2(a^2+b^2)$ we obtain

$$(\|\nabla_x\mathcal{L}(x,G\eta)\| + G|\nabla_\eta\mathcal{L}(x,G\eta)|)^2 \leq \epsilon^2$$

which completes the proof.

# B    The Stochastic Projected Gradient Descent algorithm for DRO with bounded loss

In this section we use a simple projected gradient method to minimize the DRO objective (5) and analyze its convergence rate under the assumption that the loss is bounded. Since this section is not so related to the main result in our paper, we mainly provide the gradient complexity bound in terms of $\epsilon$ for finding an $\epsilon$-stationary point without delving into problem-dependent parameters.

**Assumption B.1.** *We have $0 \leq \ell(x,\xi) \leq B$ for all $x \in \mathcal{X}$ and $\xi$.*

It turns out that we can restrict the feasible region to $\mathcal{X} \times [U,V]$ where $[U,V]$ is a finite interval.

**Proposition B.2.** *Under the Assumptions 2.4 and B.1, the DRO problem is equivalent to*

$$\text{minimize } \widehat{\mathcal{L}}(x,\eta) \qquad on \quad (x,\eta) \in \mathcal{X} \times [U,V] \tag{21}$$

*where $U = -\frac{\lambda C_\psi}{G}$ and $V = \frac{B-\lambda C_\psi}{G}$ are real numbers and $C_\psi$ is a constant depending only on $\psi$.*

**Proof:**    Note that $(\psi^*)'$ is a function satisfying the following properties:

- $(\psi^*)'$ is monotonically increasing;
- $0 \leq \lim_{s\to-\infty}(\psi^*)'(s) \leq 1$. This is because $\lim_{s\to-\infty}\frac{\psi^*(s)}{s} = \lim_{s\to-\infty}\inf_{t\geq 0}t - \frac{\psi(t)}{s} = \min\{t : \psi(t) < +\infty\} \in [0,1]$ since $\psi(1) = 0$;
- $\lim_{s\to+\infty}(\psi^*)'(s) \geq 1$ (possibly be $+\infty$). This is because $\frac{\psi^*(s)}{s} = \sup_{t\geq 0}t - \frac{\psi(t)}{s} \geq 1$ for $s > 0$ since $\psi(1) = 0$.

Therefore there exists a constant $C_\psi$ depending only on $\psi$ such that $(\psi^*)'(C_\psi) = 1$.

For any $x \in \mathcal{X}$, the optimal $\eta^*$ satisfies the following equation:

$$\mathbb{E}\left[(\psi^*)'\left(\frac{\ell(x;\xi) - G\eta^*}{\lambda}\right)\right] = 1. \tag{22}$$

We now show there exists an optimal $\eta^*$ such that $G\eta^* \in [-\lambda C_\psi, B - \lambda C_\psi]$. In fact, we have

- For any $G\eta < -\lambda C_\psi$, $\mathbb{E}\left[(\psi^*)'\left(\frac{\ell(x;\xi)-G\eta}{\lambda}\right)\right] \geq \mathbb{E}\left[(\psi^*)'\left(\frac{-\eta}{\lambda}\right)\right] \geq (\psi^*)'(C_\psi) = 1$;
- For any $G\eta > B - \lambda C_\psi$, $\mathbb{E}\left[(\psi^*)'\left(\frac{\ell(x;\xi)-G\eta}{\lambda}\right)\right] \leq \mathbb{E}\left[(\psi^*)'\left(\frac{B-\eta}{\lambda}\right)\right] \leq (\psi^*)'(C_\psi) = 1$.

We conclude the proof by noting that $\mathbb{E}\left[(\psi^*)'\left(\frac{\ell(x;\xi)-G\eta}{\lambda}\right)\right]$ is monotonically decreasing in $\eta$.    □

Since $\eta$ is constrained in a finite interval $[U,V]$, we propose to solve (21) using the randomized stochastic projected gradient (RSPG) algorithm [Ghadimi et al., 2016]. It is summarized in Algorithm 2. Note that the algorithm can deal with situations when the feasible set $\mathcal{X} \in \mathbb{R}^d$ is also constrained.

**Proposition B.3.** *Suppose Assumption 2.4 holds. Under Proposition B.2, $\mathcal{L}$ is $K$ smooth on $\mathcal{X} \times [U,V]$, where $K$ only depends on $\psi, \lambda, M, B, G$ and $L$.*

---

**Algorithm 2:** Randomized stochastic projected gradient (RSPG)

---

**Input :** Feasible region $\mathcal{K}$, objective function $F(w)$, distribution $P$, initial point $w_0 \in \mathcal{K}$, step size $\gamma$, mini-batch sizes $S$, and total number of iterations $T$

1 **for** $t \leftarrow 1$ *to* $T$ **do**
2     $\{\xi_{t-1}^{(i)}\}_{i=1}^{S} \leftarrow$ i.i.d. samples drawn from $P$;
3     $\hat{\nabla}F(w_{t-1}) \leftarrow \frac{1}{S}\sum_{i=1}^{S}\nabla F(w_{t-1}, \xi_{t-1}^{(i)})$;
4     $w_t \leftarrow \Pi_{\mathcal{K}}(w_{t-1} - \gamma\hat{\nabla}F(w_{t-1}))$ where $\Pi_{\mathcal{K}}$ is the projection onto $\mathcal{K}$;

**Output :** randomly return one $w_t$ in $\{w_t\}_{t=1}^{T}$

---

**Proof:** First note that $(\psi^*)'$ is $M$-Lipschitz continuous, and the range of $\frac{\ell(x,\xi)-G\eta}{\lambda}$ lies in the interval $\left[C_\psi - \lambda^{-1}B, C_\psi + \lambda^{-1}B\right]$, thus $(\psi^*)'\left(\frac{\ell(x;\xi)-G\eta}{\lambda}\right)$ is bounded by a constant $\left|(\psi^*)'\left(\frac{\ell(x;\xi)-G\eta}{\lambda}\right)\right| \leq (\psi^*)'(C_\psi + \lambda^{-1}B)$.

$$\|\nabla_x\mathcal{L}(x_1, \eta_1) - \nabla_x\mathcal{L}(x_2, \eta_2)\|$$
$$= \left\|\mathbb{E}_{\xi \sim P}\left[(\psi^*)'\left(\frac{\ell(x_1;\xi)-G\eta_1}{\lambda}\right)\cdot\nabla\ell(x_1,\xi) - (\psi^*)'\left(\frac{\ell(x_2;\xi)-G\eta_2}{\lambda}\right)\cdot\nabla\ell(x_2,\xi)\right]\right\|$$
$$\leq \mathbb{E}_{\xi \sim P}\left[(\psi^*)'(C_\psi + \lambda^{-1}B)\|\nabla\ell(x_1,\xi) - \nabla\ell(x_2,\xi)\|\right]$$
$$+ \mathbb{E}_{\xi \sim P}\left[G\left|(\psi^*)'\left(\frac{\ell(x_1;\xi)-G\eta_1}{\lambda}\right) - (\psi^*)'\left(\frac{\ell(x_2;\xi)-G\eta_2}{\lambda}\right)\right|\right]$$
$$\leq (\psi^*)'(C_\psi + \lambda^{-1}B)L\|x_1 - x_2\| + \lambda^{-1}GM\left(G\|x_1 - x_2\| + G\left|\eta_1 - \eta_2\right|\right)$$

Similarly we can show that

$$\|\nabla_\eta\mathcal{L}(x_1, \eta_1) - \nabla_\eta\mathcal{L}(x_2, \eta_2)\| \leq G\lambda^{-1}M\left(G\|x_1 - x_2\| + G\left|\eta_1 - \eta_2\right|\right) \tag{23}$$

Therefore $\mathcal{L}$ is smooth. $\qquad\square$

**Proposition B.4.** *Suppose Assumption 2.4 holds. Under Proposition B.2, the stochastic gradients are unbiased estimates of the true gradients $\nabla_x\mathcal{L}$ and $\nabla_\eta\mathcal{L}$ and are uniformly bounded over $\mathcal{X} \times [U, V]$, by a constant $\Lambda$ which only depends on $\psi, \lambda, M, B, G$ and $L$.*

**Proof:** As we have shown in the proof of Proposition B.3, the term $(\psi^*)'\left(\frac{\ell(x;\xi)-G\eta}{\lambda}\right)$ is bounded by $(\psi^*)'(C_\psi + \lambda^{-1}B)$. Then it's easy to see that $\nabla_x\mathcal{L}(x, \eta; \xi)$ and $\nabla_\eta\mathcal{L}(x, \eta; \xi)$ are bounded and the squared norm of true gradient is bounded by $\Lambda^2 = 2\left[(\psi^*)'(C_\psi + \lambda^{-1}B)\right]^2 G^2 + G^2$. $\qquad\square$

Following [Ghadimi et al., 2016, Reddi et al., 2016], in constrained optimization we typically consider a generalized gradient defined as

$$\mathcal{P}_\mathcal{X}(x, \nabla f(x), \gamma) = \frac{1}{\gamma}(x - x^+), \quad \text{where } x^+ = \arg\min_{u \in \mathcal{X}}\left\{\langle\nabla f(x), u\rangle + \frac{1}{2\gamma}\|u - x\|^2\right\}$$

Note that $x^+$ is exactly the projection of $x - \gamma\nabla f(x)$ onto the set $\mathcal{X}$. For unconstrained optimization when $\mathcal{X} = \mathbb{R}^d$, this definition coincides with the gradient in the traditional sense. We say that $x$ is an $\epsilon$-stationary point if $\|\mathcal{P}_\mathcal{X}(x, \nabla f(x), \gamma)\| \leq \epsilon$. The above propositions combined with [Ghadimi et al., 2016, Corollary 3] imply the following convergence result.

**Theorem B.5.** *Suppose Assumptions 2.4 and B.1 hold. With $\mathcal{K} = \mathcal{X} \times [U, V]$, $w_0 = (x_0, \eta_0)$ and properly chosen $\gamma$ and $S$, Algorithm 2 finds an $\epsilon$-stationary point with complexity $\mathcal{O}(\Lambda^2 K\Delta\epsilon^{-4})$, where $\Delta = \mathcal{L}(x_0, \eta_0) - \inf_{(x,\eta)\in\mathcal{X}\times\mathbb{R}}\mathcal{L}(x, \eta)$ and $K, \Lambda$ are constants that appeared in Propositions B.3 and B.4. Moreover, with the choice $T = 4K\Delta\epsilon^{-2}$, $\gamma = 1/2L$ and $S = 24\Lambda^2\epsilon^{-2}$, Algorithm 2 finds an $\epsilon$-stationary point with probability $\geq 0.5$.*

**Proof:** [Ghadimi et al., 2016, Corollary 3], combined with Proposition B.4 implies that if $\gamma = 1/2L$,

$$\mathbb{E}\left[\|\mathcal{P}_{\mathcal{X}\times[U,V]}((x_k, \eta_k), \nabla\mathcal{L}(x_k, \eta_k), \gamma)\|^2\right] \leq \frac{K\Delta}{T} + \frac{6\Lambda^2}{S}. \tag{24}$$

For any $\epsilon > 0$, we choose $T = 2K\Delta\epsilon^{-2}$ and $S = 12\Lambda^2\epsilon^{-2}$, then (24) implies that

$$\mathbb{E}\left[\left\|\mathcal{P}_{\mathcal{X}\times[U,V]}((x_k, \eta_k), \nabla\mathcal{L}(x_k, \eta_k), \gamma)\right\|^2\right] \leq \epsilon \tag{25}$$

Thus the sample complexity of Algorithm 1 for finding $\epsilon$-stationary point is upper bounded by $24K\Lambda^2\Delta\epsilon^{-4}$. In this case, with pobability $\geq 0.5$ the gradient norm is upper bounded by $2\epsilon$, the conclusion follows. $\qquad\square$

While the above theorem provides non-asymptotic convergence rate to a stationary point, note that the definition of generalized gradient involves the interval $[U, V]$ which was constructed artificially for Algorithm 2, thus $\mathbb{E}\left[\|\mathcal{P}_{\mathcal{X}\times[U,V]}((x_k, \eta_k), \nabla\widehat{\mathcal{L}}(x_k, \eta_k), \gamma)\|^2\right] \leq \epsilon$ does not necessarily lead to an $\epsilon$-stationary point of $\nabla\widehat{\mathcal{L}}$. We then show below that the generalized gradient is indeed equal to the true gradient in the unconstrained case $\mathcal{X} = \mathbb{R}^n$, therefore Theorem B.5 corresponds to the gradient complexity for finding an $\epsilon$-stationary point of $\Psi(x)$.

**Theorem B.6.** *Consider the unconstrained case $\mathcal{X} = \mathbb{R}^n$. Choose*

$$\tilde{U} = -\frac{\lambda C_\psi}{G} - \frac{\epsilon}{L}, \quad \tilde{V} = \frac{B - \lambda C_\psi}{G} + \frac{\epsilon}{L}$$

*as the interval constraint for $\eta$. Using parameters specified in Theorem B.5, Algorithm 2 arrives at $(x, \eta)$ with $\|\nabla\Psi(x)\| \leq \epsilon$ with probability $\geq 0.5$.*

**Proof:** It suffices to show that: whenever $\|\mathcal{P}_{\mathbb{R}^n\times[\tilde{U}, \tilde{V}]}((x, \eta), \nabla\widehat{\mathcal{L}}(x, \eta), \gamma)\| \leq \epsilon$, we must have $\|\nabla\widehat{\mathcal{L}}(x, \eta)\| \leq \epsilon$.

Recall that

$$\mathcal{P}_{\mathbb{R}^n\times[\tilde{U}, \tilde{V}]}((x, \eta), \nabla\widehat{\mathcal{L}}(x, \eta), \gamma) = \frac{1}{\gamma}(x - x^+, \eta - \eta^+) \tag{26}$$

where

$$\begin{aligned}
x^+ &= \arg\min_{u\in\mathbb{R}^n}\left\{\left\langle\nabla_x\widehat{\mathcal{L}}(x, \eta), u\right\rangle + \frac{1}{2\gamma}\|u - x\|^2\right\} \\
\eta^+ &= \arg\min_{\rho\in[\tilde{U}, \tilde{V}]}\left\{\rho\nabla_\eta\widehat{\mathcal{L}}(x, \eta) + \frac{1}{2\gamma}(\rho - \eta)^2\right\}
\end{aligned} \tag{27}$$

Define $\eta_0 := \eta - \gamma\nabla_\eta\widehat{\mathcal{L}}(x, \eta)$. Since $\|\mathcal{P}_{\mathbb{R}^n\times[\tilde{U}, \tilde{V}]}((x, \eta), \nabla\widehat{\mathcal{L}}(x, \eta), \gamma)\| \leq \epsilon$, we have $|\eta - \eta^+| \leq \gamma\epsilon$. We consider two possible cases below:

- **Case 1.** $\eta^+ \in (\tilde{U}, \tilde{V})$. In this case it is easy to see that $\eta^+ = \eta_0$ and thus

$$\|\nabla\widehat{\mathcal{L}}(x, \eta)\| = \|\mathcal{P}_{\mathbb{R}^n\times[\tilde{U}, \tilde{V}]}((x, \eta), \nabla\widehat{\mathcal{L}}(x, \eta), \gamma)\| \leq \epsilon$$

- **Case 2.** $\eta^+ \in \{\tilde{U}, \tilde{V}\}$. Assume that $\eta^+ = \tilde{U}$ (the case $\eta^+ = \tilde{V}$ is similar). Then $\eta \in [\tilde{U}, \tilde{U} + \gamma\epsilon]$. Note that $\tilde{U} + \gamma\epsilon = -\frac{\lambda C_\psi}{G} + \frac{\epsilon}{2L} < U$. In this case, $(\psi^*)'\left(\frac{\ell(x,\xi)-\eta}{\lambda}\right) \geq 1$. Therefore

$$\eta_0 = \eta - \gamma\nabla_\eta\widehat{\mathcal{L}}(x, \eta) = \eta - G\gamma\left(1 - \mathbb{E}\left[(\psi^*)'\left(\frac{\ell(x, \xi) - G\eta}{\lambda}\right)\right]\right) \geq \eta.$$

However, $\eta^+ \leq \eta$, therefore it can only be that $\eta^+ = \eta = \eta_0$. Therefore we still have

$$\|\nabla\widehat{\mathcal{L}}(x, \eta)\| = \|\mathcal{P}_{\mathbb{R}^n\times[\tilde{U}, \tilde{V}]}((x, \eta), \nabla\widehat{\mathcal{L}}(x, \eta), \gamma)\| \leq \epsilon$$

$\qquad\square$

In the above theorem, the constraint of $\eta$ is $[\tilde{U}, \tilde{V}]$ which strictly contains $\eta \in [U, V]$ (in Proposition B.2). Nevertheless, the difference of the endpoints between $U(V)$ and $\tilde{U}(\tilde{V})$ is only $\mathcal{O}(\epsilon)$. Therefore it does not change the final gradient complexity of $\mathcal{O}(\epsilon^4)$ in Theorem B.5.

# C   Proofs in Section 3.2

In this section we present the proof of main results in Section 3.2. For convenience we restate the results before proving them.

## C.1   Proofs of Lemmas 3.3 and 3.4

**Lemma C.1.** *Under Assumptions 2.4 and 3.2, the gradient estimators of* (5) *satisfies the following property:*

$$\mathbb{E}_\xi \|\nabla \widehat{\mathcal{L}}(x, \eta, \xi) - \nabla \widehat{\mathcal{L}}(x, \eta)\|^2 \leq 11 G^2 M^2 \lambda^{-2} \sigma^2 + 8(G^2 + \|\nabla \widehat{\mathcal{L}}(x, \eta)\|^2) \tag{28}$$

**Proof:**   For a random vector $X$, define the sum of its element-wise variance as

$$\mathbb{V}(X) := \mathbb{E}\|X - \mathbb{E}[X]\|_2^2, \tag{29}$$

Then it is easy to check that, for i.i.d. random vectors $X_1, X_2$ we have

$$\mathbb{E}\|X_1 - X_2\|^2 = 2\mathbb{V}[X_1]. \tag{30}$$

We first bound the variance of the stochastic gradient $\nabla_x \widehat{\mathcal{L}}(x, \eta; \xi)$. Indeed we have

$$\mathbb{V}\left[\nabla_x \widehat{\mathcal{L}}(x, \eta; \xi)\right] \tag{31}$$

$$= \frac{1}{2}\mathbb{E}_{\xi_1, \xi_2} \left\| (\psi^*)' \left( \frac{\ell(x; \xi_1) - G\eta}{\lambda} \right) \cdot \nabla \ell(x, \xi_1) - (\psi^*)' \left( \frac{\ell(x; \xi_2) - G\eta}{\lambda} \right) \cdot \nabla \ell(x, \xi_2) \right\|^2 \tag{32}$$

$$\leq \mathbb{E}_{\xi_1, \xi_2} \left[ \left( (\psi^*)' \left( \frac{\ell(x; \xi_1) - G\eta}{\lambda} \right) \right)^2 \|\nabla \ell(x, \xi_1) - \nabla \ell(x, \xi_2)\|^2 \right]$$

$$+ \mathbb{E}_{\xi_1, \xi_2} \left[ \|\nabla \ell(x, \xi_2)\|^2 \left( (\psi^*)' \left( \frac{\ell(x; \xi_1) - G\eta}{\lambda} \right) - (\psi^*)' \left( \frac{\ell(x; \xi_2) - G\eta}{\lambda} \right) \right)^2 \right] \tag{33}$$

$$\leq 4G^2 \mathbb{E}_{\xi_1} \left[ \left( (\psi^*)' \left( \frac{\ell(x; \xi_1) - G\eta}{\lambda} \right) \right)^2 \right] + G^2 M^2 \lambda^{-2} \mathbb{E}_{\xi_1, \xi_2} \left[ (\ell(x, \xi_1) - \ell(x, \xi_2))^2 \right] \tag{34}$$

$$\leq 4G^2 \mathbb{E}_{\xi_1} \left[ \left( (\psi^*)' \left( \frac{\ell(x; \xi_1) - G\eta}{\lambda} \right) \right)^2 \right] + 2G^2 M^2 \lambda^{-2} \sigma^2 \tag{35}$$

Here in (33) we use that fact that $(a + b)^2 \leq 2(a^2 + b^2)$ for any $a, b$; in (34) we use Assumption 2.4. Now we deal with the first term. Using $2(a - 1)^2 + 2 \geq a^2$ for any $a$, we have

$$\mathbb{E}_\xi \left[ \left( (\psi^*)' \left( \frac{\ell(x; \xi) - G\eta}{\lambda} \right) \right)^2 \right] \leq 2 + 2\mathbb{E}_\xi \left[ \left( 1 - (\psi^*)' \left( \frac{\ell(x; \xi) - G\eta}{\lambda} \right) \right)^2 \right]$$

$$\leq 2 \left( 1 + G^{-2}\|\nabla_\eta \widehat{\mathcal{L}}(x, \eta)\|^2 + G^{-2}\mathbb{V}[\nabla_\eta \widehat{\mathcal{L}}(x, \eta; \xi)] \right) \tag{36}$$

Next, $\mathbb{V}[\nabla_\eta \widehat{\mathcal{L}}(x, \eta; \xi)]$ can be easily bounded as follows:

$$\mathbb{V}[\nabla_\eta \widehat{\mathcal{L}}(x, \eta; \xi)] = \frac{1}{2}G^2 \mathbb{E}_{\xi_1, \xi_2} \left[ \left( (\psi^*)' \left( \frac{\ell(x; \xi_1) - G\eta}{\lambda} \right) - (\psi^*)' \left( \frac{\ell(x; \xi_2) - G\eta}{\lambda} \right) \right)^2 \right]$$

$$\leq G^2 M^2 \lambda^{-2} \sigma^2 \tag{37}$$

Combining with (35) to (37), we obtain

$$\mathbb{V}[\nabla_x \widehat{\mathcal{L}}(x, \eta; \xi)] \leq 2G^2 M^2 \lambda^{-2} \sigma^2 + 8(G^2 + \|\nabla_\eta \widehat{\mathcal{L}}(x, \eta)\|^2 + G^2 M^2 \lambda^{-2} \sigma^2)$$

$$= 10 G^2 M^2 \lambda^{-2} \sigma^2 + 8(G^2 + \|\nabla_\eta \widehat{\mathcal{L}}(x, \eta)\|^2)$$

$$\leq 10 G^2 M^2 \lambda^{-2} \sigma^2 + 8(G^2 + \|\nabla \widehat{\mathcal{L}}(x, \eta)\|^2)$$

Finally,

$$\mathbb{V}[\nabla \widehat{\mathcal{L}}(x, \eta; \xi)] = \mathbb{V}[\nabla_x \widehat{\mathcal{L}}(x, \eta; \xi)] + \mathbb{V}[\nabla_\eta \widehat{\mathcal{L}}(x, \eta; \xi)]$$

$$\leq 11 G^2 M^2 \lambda^{-2} \sigma^2 + 8(G^2 + \|\nabla \widehat{\mathcal{L}}(x, \eta)\|^2)$$

$$\square$$

**Lemma C.2.** *Under Assumption 2.4, for any pair of parameters $(x, \eta)$ and $(x', \eta')$, we have the following property for the gradient of $\widehat{\mathcal{L}}$:*

$$\|\nabla\widehat{\mathcal{L}}(x, \eta) - \nabla\widehat{\mathcal{L}}(x', \eta')\| \leq \left(K + \tfrac{L}{G}\|\nabla\widehat{\mathcal{L}}(x, \eta)\|\right)\|(x - x', \eta - \eta')\| \tag{38}$$

*where $K = L + 2G^2\lambda^{-1}M$.*

**Proof:** First write $\nabla\widehat{\mathcal{L}}(x, \eta)$ as

$$\nabla\widehat{\mathcal{L}}(x, \eta) = \mathbb{E}_\xi\left[\left((\psi^*)'\left(\frac{\ell(x; \xi) - G\eta}{\lambda}\right)\nabla\ell(x, \xi), G - G(\psi^*)'\left(\frac{\ell(x; \xi) - G\eta}{\lambda}\right)\right)^T\right] \tag{39}$$

We then split $\nabla\mathcal{L}(x, \eta) - \nabla\mathcal{L}(x', \eta')$ into two terms $A + B$, where

$$\begin{aligned}
A &= \mathbb{E}_\xi\left[\left((\psi^*)'\left(\frac{\ell(x; \xi) - G\eta}{\lambda}\right)(\nabla\ell(x; \xi) - \nabla\ell(x'; \xi)), 0\right)^T\right] \\
B &= \mathbb{E}_\xi\left[\left((\psi^*)'\left(\frac{\ell(x; \xi) - G\eta}{\lambda}\right) - (\psi^*)'\left(\frac{\ell(x'; \xi) - G\eta'}{\lambda}\right)\right)(\nabla\ell(x'; \xi), -G)^T\right].
\end{aligned} \tag{40}$$

$A$ can be bounded as follows:

$$\|A\| \leq L \cdot \mathbb{E}_\xi\left[(\psi^*)'\left(\frac{\ell(x; \xi) - G\eta}{\lambda}\right)\|x - x'\|\right] \tag{41}$$

where we use $(\psi^*)'(s) \geq 0$ for all $s$. $B$ can be bounded as follows:

$$\begin{aligned}
\|B\| &\leq \sqrt{2}G \cdot \mathbb{E}_\xi\left[\left|(\psi^*)'\left(\frac{\ell(x; \xi) - G\eta}{\lambda}\right) - (\psi^*)'\left(\frac{\ell(x'; \xi) - G\eta'}{\lambda}\right)\right|\right] \\
&\leq \sqrt{2}G\lambda^{-1}M\mathbb{E}_\xi\left[|(\ell(x; \xi) - \ell(x'; \xi)) - G(\eta - \eta')|\right] \\
&\leq 2G^2\lambda^{-1}M\|(x, \eta)^T - (x', \eta')^T\|
\end{aligned} \tag{42}$$

where the last step is because the function $(x, \eta) \to \ell(x, \xi) - G\eta$ is $\sqrt{2}G$ Lipschitz. Finally we bound $\mathbb{E}_\xi\left[(\psi^*)'\left(\frac{\ell(x;\xi) - G\eta}{\lambda}\right)\right]$ using the true gradient of $\widehat{\mathcal{L}}$:

$$\mathbb{E}_\xi\left[(\psi^*)'\left(\frac{\ell(x; \xi) - G\eta}{\lambda}\right)\right] = 1 - G^{-1}\nabla_\eta\widehat{\mathcal{L}}(x, \eta) \leq 1 + G^{-1}|\nabla_\eta\widehat{\mathcal{L}}(x, \eta)|$$

Combining the above inequalities, we obtain

$$\begin{aligned}
\|\nabla\widehat{\mathcal{L}}(x, \eta) - \nabla\widehat{\mathcal{L}}(x', \eta')\| &\leq \|A\| + \|B\| \\
&\leq (L + LG^{-1})|\nabla_\eta\widehat{\mathcal{L}}(x, \eta)|\|x - x'\| + 2G^2\lambda^{-1}M\|(x - x', \eta - \eta')^T\| \\
&\leq \left(L + 2G^2\lambda^{-1}M + \tfrac{L}{G}\|\nabla\widehat{\mathcal{L}}(x, \eta)\|\right)\|(x - x', \eta - \eta')\|
\end{aligned}$$

which concludes the proof. $\qquad\qquad\square$

## C.2 Proof of Theorem 3.5

### C.2.1 Properties of generalized smoothness

We formalize the generalized smoothness property into a definition.

**Definition C.3.** *A continuously differentiable function $F : \mathbb{R}^d \to \mathbb{R}$ is said to be $(K_0, K_1)$-smooth if $\|\nabla F(x) - \nabla F(y)\| \leq (K_0 + K_1\|\nabla F(x)\|)\|x - y\|$ for all $x, y \in \mathbb{R}^d$.*

We now present a descent inequality for $(K_0, K_1)$-smooth functions which will be used in subsequent analysis.

**Lemma C.4.** *(Descent Inequality) Let $F$ be $(K_0, K_1)$-smooth, then for any point $x$ and direction $z$ the following holds:*

$$F(x - z) \leq F(x) - \langle\nabla F(x), z\rangle + \frac{1}{2}(K_0 + K_1\|\nabla F(x)\|)\|z\|^2. \tag{43}$$

**Proof:** By definition we have

$$
\begin{aligned}
F(x-z) - F(x) - \langle z, \nabla F(x) \rangle &= \int_0^1 \langle \nabla F(x - \theta z) - \nabla F(x), z \rangle \, \mathrm{d}\theta \\
&\leq \int_0^1 \|\nabla F(x - \theta z) - \nabla F(x)\| \|z\| \mathrm{d}\theta \\
&\leq \int_0^1 \left( K_0 \theta \|z\|^2 + K_1 \theta \|z\|^2 \|\nabla F(x)\| \right) \mathrm{d}\theta \\
&= \frac{K_0 + K_1 \|\nabla F(x)\|}{2} \|z\|^2
\end{aligned}
\tag{44}
$$

so the conclusion follows. $\qquad\square$

### C.2.2 Properties of the normalized update

We begin with a simple algebraic lemma.

**Lemma C.5.** *Let $\mu \geq 0$ be a real constant. For any vectors $u$ and $v$,*

$$
-\frac{\langle u, v \rangle}{\|v\|} \leq -\mu \|u\| - (1 - \mu)\|v\| + (1 + \mu)\|v - u\|
\tag{45}
$$

**Proof:**

$$
\begin{aligned}
-\frac{\langle u, v \rangle}{\|v\|} &= -\|v\| + \frac{\langle v - u, v \rangle}{\|v\|} \\
&\leq -\|v\| + \|v - u\| \\
&\leq -\|v\| + \|v - u\| + \mu(\|v - u\| + \|v\| - \|u\|) \\
&= -\mu \|u\| - (1 - \mu)\|v\| + (1 + \mu)\|v - u\|
\end{aligned}
$$

$\qquad\square$

Now we can characterize the behavior of normalization-based algorithms in terms of function value descent.

**Lemma C.6.** *Consider the algorithm that starts at $w_0$ and makes updates $w_{t+1} = w_t - \gamma \frac{m_{t+1}}{\|m_{t+1}\|}$ where $\{m_t\}$ is an arbitrary sequence of points. Define $\delta_t := m_{t+1} - \nabla F(w_t)$ be the estimation error. Then*

$$
F(w_{t+1}) - F(w_t) \leq -\left( \gamma - \frac{1}{2} K_1 \gamma^2 \right) \|\nabla F(w_t)\| + \frac{1}{2} K_0 \gamma^2 + 2\gamma \|\delta_t\|
$$

*And thus by a telescope sum we have*

$$
\left( 1 - \frac{1}{2} K_1 \gamma \right) \sum_{t=0}^{T-1} \|\nabla F(w_t)\| \leq \frac{F(w_0) - F(w_T)}{\gamma} + \frac{1}{2} K_0 T \gamma + 2 \sum_{t=0}^{T-1} \|\delta_t\|
$$

**Proof:** Since $\|w_{t+1} - w_t\| = \gamma$, by Lemma C.4 we have

$$
\begin{aligned}
F(w_{t+1}) - F(w_t) &\leq -\frac{\gamma}{\|m_{t+1}\|} \langle \nabla F(w_t), m_{t+1} \rangle + \frac{1}{2} \gamma^2 \left( K_0 + K_1 \|\nabla F(w_t)\| \right) \\
&\leq \gamma \left( -\|\nabla F(w_t)\| + 2\|\delta_t\| \right) + \frac{1}{2} \gamma^2 \left( K_0 + K_1 \|\nabla F(w_t)\| \right) \\
&= -\left( \gamma - \frac{1}{2} K_1 \gamma^2 \right) \|\nabla F(w_t)\| + \frac{1}{2} K_0 \gamma^2 + 2\gamma \|\delta_t\|
\end{aligned}
$$

where in the second inequality we use Lemma C.5. $\qquad\square$

### C.2.3 A general convergence result

Instead of directly focusing on the specific problem of DRO, we first provide convergence guarantee for Algorithm 2 under general smoothness and noise assumptions.

**Theorem C.7.** *Suppose that $F$ is $(K_0, K_1)$-smooth and the stochastic gradient estimator $\hat{\nabla} F(w, \xi)$ is unbiased and satisfies*

$$\mathbb{E} \left\| \hat{\nabla} F(w, \xi) - \nabla F(w) \right\|^2 \leq \Gamma^2 \left\| \nabla F(w) \right\|^2 + \Lambda^2$$

*Let $\{w_t\}$ be the sequence produced by Algorithm 1, then with a mini-batch size $S = 64\Gamma^2$ and a suitable choice of parameters $\gamma$ and $\beta$, for any small $\epsilon \leq \min \left( \frac{K_0}{K_1}, \frac{\Lambda}{2\Gamma} \right)$, we need at most $512 \Delta K_0 \Lambda^2 \epsilon^{-4}$ gradient complexity to guarantee that we find an $2\epsilon$-first-order stationary point in expectation, i.e. $\frac{1}{T} \sum_{t=0}^{T-1} \mathbb{E} \|\nabla F(w_t)\| \leq 2\epsilon$ where $\Delta = F(w_0) - \inf_{w \in \mathbb{R}^d} F(w)$.*

**Proof:** Define the estimation errors $\delta_t := m_{t+1} - \nabla F(w_t)$. Denote $H(a, b) := \nabla F(a) - \nabla F(b)$. We can upper bound $H(a, b)$ using the definition of $(K_0, K_1)$-smoothness:

$$\|H(a, b)\| \leq \|a - b\| \left( K_0 + K_1 \|\nabla F(a)\| \right) \tag{46}$$

Using the definition of momentum $m_t$ and $H(a, b)$, we can get a recursive formula on $\delta_t$:

$$\begin{aligned}
\delta_{t+1} &= \beta m_{t+1} + (1 - \beta) \hat{\nabla} F(w_{t+1}) - \nabla F(w_{t+1}) \\
&= \beta \delta_t + \beta H(w_t, w_{t+1}) + (1 - \beta)(\hat{\nabla} F(w_{t+1}) - \nabla F(w_{t+1}))
\end{aligned} \tag{47}$$

Denote $\hat{\delta}_t = \hat{\nabla} F(w_t) - \nabla F(w_t)$ be the stochastic noise, then the variance of $\hat{\delta}_t$ can be bounded by $\mathbb{E} \|\hat{\delta}_t\|^2 \leq \frac{1}{S} \left( \Gamma^2 \|\nabla F(w_t)\|^2 + \Lambda^2 \right)$. After applying (47) recursively and plugging $\hat{\delta}_t$ into (47) we obtain

$$\delta_t = \beta \sum_{\tau=0}^{t-1} \beta^\tau H(w_{t-\tau-1}, w_{t-\tau}) + (1 - \beta) \sum_{\tau=0}^{t-1} \beta^\tau \hat{\delta}_{t-\tau} + (1 - \beta)\beta^t \hat{\delta}_0 + \beta^{t+1}(m_0 - \nabla F(w_0))$$

Using triangle inequality and plugging in the estimate (46), we have

$$\|\delta_t\| \leq (1 - \beta) \left\| \sum_{\tau=0}^t \beta^\tau \hat{\delta}_{t-\tau} \right\| + \beta \gamma \sum_{\tau=0}^{t-1} \beta^\tau \left( K_0 + K_1 \|\nabla F(w_{t-\tau-1})\| \right) + \beta^{t+1} \|m_0 - \nabla F(w_0)\| \tag{48}$$

Taking a telescope summation of (48) we obtain

$$\sum_{t=0}^{T-1} \|\delta_t\| \leq (1 - \beta) \sum_{t=0}^{T-1} \left\| \sum_{\tau=0}^t \beta^\tau \hat{\delta}_{t-\tau} \right\| + \frac{K_0 T \gamma \beta}{1 - \beta} + \frac{K_1 \gamma \beta}{1 - \beta} \sum_{t=0}^{T-1} \|\nabla F(w_t)\| + \frac{\beta}{1 - \beta} \|m_0 - \nabla F(w_0)\| \tag{49}$$

Now we take expectation of $\left\| \sum_{\tau=0}^t \beta^\tau \hat{\delta}_{t-\tau} \right\|$ over all the randomness. We will prove a core lemma (Lemma C.9) later which shows

$$\mathbb{E} \left\| \sum_{\tau=0}^t \beta^\tau \hat{\delta}_{t-\tau} \right\| \leq \frac{\Lambda}{\sqrt{(1 - \beta^2)S}} + \frac{\Gamma}{\sqrt{S}} \sum_{\tau=0}^t \beta^\tau \mathbb{E}[\|\nabla F(w_{t-\tau})\|] \tag{50}$$

Now substituting (50) into (49) we obtain

$$\begin{aligned}
\mathbb{E} \left[ \sum_{t=0}^{T-1} \|\delta_t\| \right] \leq &\frac{K_0 T \gamma \beta}{1 - \beta} + \frac{K_1 \gamma \beta}{1 - \beta} \sum_{t=0}^{T-1} \mathbb{E} \|\nabla F(w_t)\| + \frac{\beta}{1 - \beta} \|m_0 - \nabla F(w_0)\| \\
&+ \frac{\Lambda T \sqrt{1 - \beta}}{\sqrt{S}} + \frac{\Gamma}{\sqrt{S}} \sum_{t=0}^{T-1} \mathbb{E} \|\nabla F(w_t)\|
\end{aligned} \tag{51}$$

Finally we substitute (51) into Lemma C.6:

$$
\left(1 - \left(\frac{1}{2} + \frac{2\beta}{1-\beta}\right)K_1\gamma - \frac{2\Gamma}{\sqrt{S}}\right)\mathbb{E}\sum_{t=0}^{T-1}\|\nabla F(w_t)\|
$$

$$
\leq \frac{\Delta}{\gamma} + \frac{1}{2}K_0T\gamma + 2\left(\frac{\sqrt{1-\beta}T\Lambda}{\sqrt{S}} + \frac{K_0T\gamma\beta}{1-\beta} + \frac{\beta}{1-\beta}\|m_0 - \nabla F(w_0)\|\right)
$$

If we choose $\gamma = \frac{1}{8}(\min(K_1^{-1}, K_0^{-1}\epsilon)(1-\beta)$, and $S = 64\Gamma^2$, then

$$
\left(1 - \left(\frac{1}{2} + \frac{2\beta}{1-\beta}\right)K_1\gamma\right) - \frac{2\Gamma}{\sqrt{S}} = \left(1 - \frac{1+3\beta}{2(1-\beta)}K_1\gamma\right) - \frac{1}{4} \geq \frac{3}{4} - \frac{2K_1\gamma}{1-\beta} \geq \frac{1}{2}
$$

In this case

$$
\frac{1}{T}\mathbb{E}\sum_{t=0}^{T-1}\|\nabla F(w_t)\| \leq 2\left(\frac{\Delta}{\gamma T} + \frac{1}{2}K_0\gamma + \frac{2K_0\gamma\beta}{1-\beta} + \frac{\sqrt{1-\beta}\Lambda}{4\Gamma} + \frac{2\beta}{(1-\beta)T}\|m_0 - \nabla F(w_0)\|\right)
$$

$$
\leq 2\left(\frac{\Delta}{\gamma T} + \frac{1}{4}\epsilon + \frac{\sqrt{1-\beta}\Lambda}{4\Gamma} + \frac{2\beta}{(1-\beta)T}\|m_0 - \nabla F(w_0)\|\right)
$$

Set $1 - \beta = \min(4\Lambda^{-2}\Gamma^2\epsilon^2, 1)$ and $m_0 = \|\nabla F(w_0)\|$, then

$$
\frac{1}{T}\mathbb{E}\sum_{t=0}^{T-1}\|\nabla F(w_t)\| \leq \frac{3}{2}\epsilon + \frac{2\Delta}{\gamma T}
$$

Therefore for $T = \frac{4\Delta}{\gamma\epsilon}$, we have $\frac{1}{T}\mathbb{E}\sum_{t=0}^{T-1}\|\nabla F(w_t)\| \leq 2\epsilon$. The total gradient complexity is

$$
ST = \frac{2048\Gamma^2\Delta\max(K_1, K_0\epsilon^{-1})}{\min(4\Gamma^2\Lambda^{-2}\epsilon^2, 1)\epsilon}.
$$

If $\epsilon \leq \min\left(\frac{K_0}{K_1}, \frac{\Lambda}{2\Gamma}\right)$, then the gradient complexity is $512\Lambda^2\Delta K_0\epsilon^{-4}$. $\qquad\square$

**Corollary C.8.** *Suppose the DRO problem* (3) *satisfies Assumptions 2.4 and 3.2. Using Algorithm 1 with a constant batch size 4096, the gradient complexity for finding an $\epsilon$-stationary point of $\Psi(x)$ is*

$$
\mathcal{O}\left(G^2\left(M^2\sigma^2\lambda^{-2} + 1\right)\left(\lambda^{-1}MG^2 + L\right)\Delta\epsilon^{-4}\right).
$$

**Proof:** Lemmas 3.3 and 3.4 imply that the conditions in Theorem 3.5 for $\widehat{\mathcal{L}}(x, \eta)$ are satisfied with $K_0 = L + 2G^2\lambda^{-1}M, \Gamma^2 = 64, \Lambda^2 = 11G^2M^2\lambda^{-2}\sigma^2 + 8G^2$. The main result immediately follows from Theorems 2.7 and 3.5. $\qquad\square$

We now return to prove the core lemma that is used in (50).

**Lemma C.9.** *Let $\hat{\delta}_t = \hat{\nabla}F(w_t) - \nabla F(w_t)$ be the stochastic noise. Then*

$$
\mathbb{E}\left\|\sum_{\tau=0}^{t}\beta^\tau\hat{\delta}_{t-\tau}\right\| \leq \frac{\Lambda}{\sqrt{(1-\beta^2)S}} + \frac{\Gamma}{\sqrt{S}}\sum_{\tau=0}^{t}\beta^\tau\mathbb{E}[\|\nabla F(w_{t-\tau})\|]. \tag{52}
$$

**Proof:** We prove the following result: for each $i \in \{0, 1, \cdots, t+1\}$, the following inequality holds:

$$
\mathbb{E}\left\|\sum_{\tau=0}^{t}\beta^\tau\hat{\delta}_{t-\tau}\right\| \leq \frac{\Gamma}{\sqrt{S}}\sum_{\tau=t-i+1}^{t}\beta^{t-\tau}\mathbb{E}\|\nabla F(w_\tau)\| + \mathbb{E}\left[\sqrt{\frac{\Lambda^2}{S}\sum_{\tau=t-i+1}^{t}\beta^{2(t-\tau)} + \left\|\sum_{\tau=i}^{t}\beta^\tau\hat{\delta}_{t-\tau}\right\|^2}\right]. \tag{53}
$$

It is easy to see that Lemma C.9 follows by setting $i = t+1$ in (53).

We prove (53) by induction. When $i = 0$, (53) holds obviously. Now suppose (53) holds for $i$, and we want to prove that (53) holds for $i + 1$.

Let $\mathcal{F}_t$ denote the $\sigma$-algebra generated by the stochastic gradient noise in the first $t$ iterations, i.e. $\{\xi_\tau^{(i)} : i \in \{1, \cdots, S\}, \tau \in \{0, \cdots, t\}\}$ in Algorithm 1. We use $\mathbb{E}_t$ to denote the conditional expectation on $\mathcal{F}_t$. In other words, $\mathbb{E}_t$ takes expectation over the randomness in subsequent $T - t$ iterations after the first $t$ iterations finish and become deterministic. We also use $\mathbb{E}_{\mathcal{F}_t}$ to denote the expectation on $\mathcal{F}_t$. We have

$$\mathbb{E}\left[\sqrt{\frac{\Lambda^2}{S}\sum_{\tau=t-i+1}^{t}\beta^{2(t-\tau)} + \left\|\sum_{\tau=i}^{t}\beta^\tau\hat{\delta}_{t-\tau}\right\|^2}\right] \tag{54}$$

$$= \mathbb{E}_{\mathcal{F}_{t-i-1}}\left[\mathbb{E}_{t-i-1}\left[\sqrt{\frac{\Lambda^2}{S}\sum_{\tau=t-i+1}^{t}\beta^{2(t-\tau)} + \left\|\sum_{\tau=i}^{t}\beta^\tau\hat{\delta}_{t-\tau}\right\|^2}\right]\right] \tag{55}$$

$$\leq \mathbb{E}_{\mathcal{F}_{t-i-1}}\left[\sqrt{\mathbb{E}_{t-i-1}\left[\frac{\Lambda^2}{S}\sum_{\tau=t-i+1}^{t}\beta^{2(t-\tau)} + \left\|\sum_{\tau=i}^{t}\beta^\tau\hat{\delta}_{t-\tau}\right\|^2\right]}\right] \tag{56}$$

$$\leq \mathbb{E}_{\mathcal{F}_{t-i-1}}\left[\sqrt{\mathbb{E}_{t-i-1}\left[\frac{\Lambda^2}{S}\sum_{\tau=t-i+1}^{t}\beta^{2(t-\tau)} + \beta^{2i}\|\hat{\delta}_{t-i}\|^2 + \left\|\sum_{\tau=i+1}^{t}\beta^\tau\hat{\delta}_{t-\tau}\right\|^2\right]}\right] \tag{57}$$

$$\leq \mathbb{E}_{\mathcal{F}_{t-i-1}}\left[\sqrt{\mathbb{E}_{t-i-1}\left[\frac{\Lambda^2}{S}\sum_{\tau=t-i+1}^{t}\beta^{2(t-\tau)} + \frac{\beta^{2i}}{S}(\Gamma^2\|\nabla F(w_{t-i})\|^2 + \Lambda^2) + \left\|\sum_{\tau=i+1}^{t}\beta^\tau\hat{\delta}_{t-\tau}\right\|^2\right]}\right] \tag{58}$$

$$= \mathbb{E}_{\mathcal{F}_{t-i-1}}\left[\sqrt{\frac{\beta^{2i}}{S}\Gamma^2\|\nabla F(w_{t-i})\|^2 + \frac{\Lambda^2}{S}\sum_{\tau=t-i}^{t}\beta^{2(t-\tau)} + \left\|\sum_{\tau=i+1}^{t}\beta^\tau\hat{\delta}_{t-\tau}\right\|^2}\right] \tag{59}$$

$$\leq \mathbb{E}_{\mathcal{F}_{t-i-1}}\left[\frac{\beta^i}{\sqrt{S}}\Gamma\|\nabla F(w_{t-i})\| + \sqrt{\frac{\Lambda^2}{S}\sum_{\tau=t-i}^{t}\beta^{2(t-\tau)} + \left\|\sum_{\tau=i+1}^{t}\beta^\tau\hat{\delta}_{t-\tau}\right\|^2}\right] \tag{60}$$

$$= \frac{\beta^i}{\sqrt{S}}\Gamma\mathbb{E}\left[\|\nabla F(w_{t-i})\|\right] + \mathbb{E}\left[\sqrt{\frac{\Lambda^2}{S}\sum_{\tau=t-i}^{t}\beta^{2(t-\tau)} + \left\|\sum_{\tau=i+1}^{t}\beta^\tau\hat{\delta}_{t-\tau}\right\|^2}\right] \tag{61}$$

Here in (55) we use the property of conditional expectation; In (56) we use $\mathbb{E}[X^2] \geq (\mathbb{E}[X])^2$ for any random variable $X$; In (57) we use the fact that $\hat{\delta}_\tau, \tau < t$ are $\mathcal{F}_{t-1}$-measurable, and are uncorrelated with $\hat{\delta}_t$; In (58) we use the noise assumption; In (59) we use the fact that $w_{t-i}$ is $\mathcal{F}_{t-i-1}$-measurable; In (60) we use the fact that $\sqrt{a+b} \leq \sqrt{a} + \sqrt{b}$ for all $a \geq 0, b \geq 0$. Proof completed. $\qquad\square$

# D   Proofs in Section 3.4

In this section we prove the main result of Section 3.4 for smoothed CVaR. Recall the expressions

$$\psi_\alpha^{\text{smo}}(t) = \begin{cases} t\log t + \frac{1-\alpha t}{\alpha}\log\frac{1-\alpha t}{1-\alpha} & t \in [0, 1/\alpha) \\ +\infty & \text{otherwise} \end{cases} \tag{62}$$

$$\psi_\alpha^{\text{smo},*}(t) = \frac{1}{\alpha}\log(1 - \alpha + \alpha\exp(t)). \tag{63}$$

The following proposition shows that $\psi_\alpha^{\text{smo},*}$ is Lipschitz-continuous and smooth.

**Proposition D.1.** $\psi_\alpha^{\text{smo},*}(t)$ is $\frac{1}{\alpha}$-Lipschitz and $\frac{1}{4\alpha}$-smooth.

**Proof:** We have

$$\left(\psi_\alpha^{\text{smo},*}\right)'(t) = \frac{1}{\alpha}\frac{\alpha\exp(t)}{1-\alpha+\alpha\exp(t)} \leq \frac{1}{\alpha}, \tag{64}$$

$$\left(\psi_\alpha^{\text{smo},*}\right)''(t) = \frac{1}{\alpha}\frac{\alpha(1-\alpha)\exp(t)}{(1-\alpha+\alpha\exp(t))^2} \leq \frac{1}{4\alpha}. \tag{65}$$

where we use $\alpha(1-\alpha) \leq \frac{1}{4}$. Hence the conclusion follows. $\qquad\square$

**Proposition D.2.** *Fix $0 < \alpha < 1$. When $\lambda \to 0$, the solution of the DRO problem* (5) *for smoothed CVaR tends to the solution for the standard CVaR.*

**Proof:** For the standard CVaR, the DRO problem can be written as

$$\mathcal{L}^{\text{CVaR}}(x,\eta) := \lambda\mathbb{E}_\xi\left[\max\left(\frac{\ell(x;\xi)-\eta}{\alpha\lambda}, 0\right)\right] + \eta = \frac{1}{\alpha}\mathbb{E}_\xi\left[\max\left(\ell(x;\xi)-\eta, 0\right)\right] + \eta \tag{66}$$

which is irrelevant to $\lambda$. For smoothed CVaR, the DRO problem can be written as

$$\mathcal{L}_\lambda^{\text{SCVaR}}(x,\eta) := \frac{\lambda}{\alpha}\mathbb{E}_\xi\left[\log\left(1-\alpha+\alpha\exp\left(\frac{\ell(x;\xi)-\eta}{\lambda}\right)\right)\right] + \eta \tag{67}$$

It is easy to see that $\lim_{\lambda\to 0^+}\lambda\log\left(1-\alpha+\alpha\exp\left(\frac{z}{\lambda}\right)\right) = \max(z,0)$ for any $z \in \mathbb{R}$. Therefore (67) tends to (66) when $\lambda \to 0^+$. $\qquad\square$

**Lemma D.3.** *Suppose Assumption 2.4 holds. For smoothed CVaR, the DRO objective* (5) *satisfies*

$$\mathbb{E}\|\nabla\widehat{\mathcal{L}}(x,\eta,\xi)\|^2 \leq 2\alpha^{-2}G^2. \tag{68}$$

*Moreover, $\widehat{\mathcal{L}}(x,\eta)$ is $K$-smooth with $K = \frac{L}{\alpha} + \frac{G^2}{2\lambda\alpha}$.*

**Proof:** We have

$$\|\nabla_x\widehat{\mathcal{L}}(x,\eta;\xi)\| = (\psi^*)'\left(\frac{\ell(x,\xi)-G\eta}{\lambda}\right)\|\nabla\ell(x;\xi)\|$$

$$\leq \alpha^{-1}\|\nabla\ell(x;\xi)\| \leq \alpha^{-1}G$$

since $\psi^*$ is non-decreasing and $\frac{1}{\alpha}$-Lipschitz continuous.

We also have $\left\|\nabla_\eta\widehat{\mathcal{L}}(x,\eta;\xi)\right\| \leq \alpha^{-1}G$. Therefore $\left\|\nabla\widehat{\mathcal{L}}(x,\eta)\right\|^2 \leq 2\alpha^{-2}G^2$.

Now we turn to the smoothness of $\mathcal{L}$. For any $(x,\eta)$ and $(x',\eta')$ we decouple $\nabla\widehat{\mathcal{L}}(x,\eta) - \nabla\widehat{\mathcal{L}}(x',\eta')$ into $A + B$ using the same approach as in (40). Now different from (41), $A$ can be bounded by

$$\|A\| \leq \frac{L}{\alpha}\|x - x'\| \tag{69}$$

using the Lipschitz property of $\psi^*$. The bound for $B$ is the same as (42):

$$\|B\| \leq \frac{G^2}{2\lambda\alpha}\|(x,\eta)^T - (x',\eta')^T\| \tag{70}$$

Hence $\mathcal{L}$ is $K$-smooth as desired. $\qquad\square$

**Theorem D.4.** *Suppose that $\psi = \psi_\alpha^{smo}$ and Assumption 2.4 holds. If we run SGD with properly selected hyper-parameters on the loss $\widehat{\mathcal{L}}(x,\eta)$, then the gradient complexity of finding an $\epsilon$-stationary point of $\Psi(x)$ is $\mathcal{O}\left(\alpha^{-3}\lambda^{-1}G^2(G^2+\lambda L)\Delta\epsilon^{-4}\right)$, where $\Delta = \mathcal{L}(x_0,\eta_0) - \inf_x\Psi(x)$.*

**Proof:** It is well-known [Ghadimi and Lan, 2013] that the complexity of SGD for finding an $\epsilon$-stationary point is $\mathcal{O}(\Delta K\sigma^2\epsilon^{-4})$ if the objective function is $K$-smooth and $\sigma^2$ is an upper bound of the variance of stochastic gradients. Now the proof can be completed by using Lemma D.3. $\qquad\square$

# E Experiment

## E.1 Dataset description

**Imbalanced CIFAR-10.** To demonstrate the effectiveness of our method in DRO-classification setting, we construction an imbalanced classification dataset. The original version of CIFAR-10 contains 50,000 training images and 10,000 validation images of size 32×32 with 10. To create their imbalanced version, we reduce the number of training examples per class and keep the validation set unchanged. We consider the type of random imbalance and use $\rho_i$ to denote the sample ratio of $i$th class between the imbalanced and original dataset. $\rho = \{0.804, 0.543, 0.997, 0.593, 0.390, 0.285, 0.959, 0.806, 0.967, 0.660\}$

## E.2 Implementation details

For every training task we jointly tune the parameters learning rate for baseline and our method by grid search and pick the one that achieves the fastest optimization. By default we set momentum = 0.9 for all experiments and $\epsilon = 0.1$ for normalized SGD. We use batch size n = 128 throughout.

**Hyper-parameter for $\chi^2$ penalized DRO.** In regression setting, we use SGD with lr=0.0002 as our baseline algorithm and set lr=0.005 for normalized SGD. In classification setting, we set lr=0.005 and 0.01 for baseline and our method, respectively.

**Hyper-parameter for smoothed CVaR.** In smooth CVaR, we also divide experiment into two part, regression and classification task. We train CVaR with lr = (0.00005, 0.00005) and smoothed CVaR with lr = (0.001, 0.0001) in regression and classification setting.

Table 3: Test performance of CVaR-DRO problem for unbalanced CIFAR-10 classification. Each column corresponds to the performance of a particular class. The bolded column indicates the worst-performing class.

| Class | 1 | 2 | 3 | 4 | 5 | **6** | 7 | 8 | 9 | 10 |
|---|---|---|---|---|---|---|---|---|---|---|
| Number of training samples | 4020 | 2715 | 4985 | 2965 | 1950 | **1425** | 4795 | 4030 | 4835 | 3300 |
| Test acc (CVaR) | 63.0 | 52.6 | 57.9 | 36.2 | 42.1 | **35.4** | 67.4 | 59.1 | 80.9 | 60.6 |
| Test acc (Smoothed CVaR) | 74.6 | 73.6 | 67.8 | 50.3 | 53.1 | **37.2** | 80.2 | 79.3 | 90.2 | 67.1 |