# OpenReview forum: "Non-convex Distributionally Robust Optimization: Non-asymptotic Analysis"
_NeurIPS.cc/2021/Conference — NeurIPS 2021 Poster_

### Official Review · Reviewer_Fpn4 · 2021-07-16

**Rating:** 7
**Confidence:** 3

**Summary:**

This paper studies distributionally robust optimization with the loss function being nonconvex and the uncertainty set defined by a divergence function. Particularly, this paper gives a very interesting generalized smoothness condition, shows the connection of the normalized gradient descent with this smoothness condition, and gives a non-asymptotic convergence rate. Finally, experimental results verify the theory.

**Limitations And Societal Impact:**

No Societal Impact

**Main Review:**

I believe that this is a good paper in this area. Both the theoretical and experimental results are good. My main concern is that this paper can only address the soft constrained case. For the CVaR case or the hard constrained case, can we still apply the technique in this paper, such as the generalized smoothness condition? For instance, for the hard constrained case, we can get a similar reformulation about $(x,\eta)$ and $\lambda$ (see Namkoong & Duchi, 2016).  When $\lambda$ is also changing, can the generalized smoothness condition still hold?

The O(\epsilon^{-4}) may be further improved if we use some variance reduction approach based on SARAH/SPIDER.

Does the mini-batch setting really necessary? Can we only use one sample and get the same rate by decreasing the step size?


Some minor comments:

1. In Lemma 3.4, we may use $x', \eta'$ to replace $x_2, \eta_2$.


**Time Spent Reviewing:**

5

---

> ### Author Response · Authors · 2021-08-09
> **Response to Reviewer Fpn4**
>
> We thank the reviewer for the thoughtful review and inspiring suggestions. Below are our responses to the questions and suggestions raised by the reviewer.
>
> **About the hard constraint setting.** As the reviewer suggests, Namkoong \& Duchi (2016) pointed out that SGD fails to converge in the hard constraint setting, mainly because the DRO objective would have exploding gradients if $\lambda \to 0$. We think that similar issue may occur in our approach, so that our method cannot be directly applied to the hard constraint setting. However, we think that our method can probably be combined with some other techniques (e.g. the MLMC estimator used in [1] ) to handle the hard constraint problem.
>
> **Variance reduction methods.** We thank the reviewer for suggesting the variance reduction approaches to achieve acceleration. We are aware that methods such as SPIDER also use gradient normalization, so it might be possible to apply these methods in our setting. We think this is an interesting and important future direction.
>
> **About the use of mini-batch in our method.** In this paper we use a mini-batch at each iteration to reduce the variance of the stochastic gradient estimator. We think the use of mini-batch is necessary because the performance of normalization-based methods crucially depends on the quality of the stochastic gradient estimators. For example, it is known that normalized SGD (without momentum) fails to converge unless a large batch size (which depends on the accuracy $\epsilon$) is used [2].
>
> The use of momentum in the update also plays a role of variance reduction, and it suffices to use one sample at each iteration when the gradient of the objective function has bounded variance. However, in our setting the stochastic gradient variance depends on the magnitude of the true gradient, and we find that such dependence can only be reduced by using mini-batch rather than momentum (cf. Theorem C.7 in the Appendix), otherwise the descent of a step (cf. Lemma 3.8) would be overridden by the noise of the gradient estimator. In particular we think descent cannot be ensured by tuning the step size because in the 'descent inequality' of Lemma 3.8, both the gradient term and the error term has coefficient $\mathcal{O}(\gamma)$ (where $\gamma$ is the step size). This is a special feature of normalized update and is different from the analysis SGD where the error term has coefficient $\mathcal{O}(\gamma^2)$.
>
>
> [1]Levy, D., Carmon, Y., Duchi, J. C.,\& Sidford, A. (2020). Large-Scale Methods for Distributionally Robust Optimization. Advances in Neural Information Processing Systems, 33.
>
> [2]Hazan, E., Levy, K., \& Shalev-Shwartz, S. (2015). Beyond Convexity: Stochastic Quasi-Convex Optimization. Advances in Neural Information Processing Systems, 28, 1594-1602.

---

### Official Review · Reviewer_u9sJ · 2021-07-16

**Rating:** 8
**Confidence:** 5

**Summary:**

This paper bridges the gap by studying DRO algorithms for general smooth non-convex losses. By carefully exploiting the specific form of the DRO objective with $\psi$ divergence, this paper provides non-asymptotic convergence guarantees even though the objective function is possibly non-convex, non-smooth, and has unbounded gradient noise. In particular, the authors prove that a special algorithm called mini-batch normalized gradient descent with momentum can find an $\epsilon$-first-order stationary point within $O(\epsilon^{-4})$ gradient complexity. Furthermore, for the conditional value-at-risk (CVaR) setting, this paper proposes a penalized DRO objective based on a smoothed version of the CVaR and obtains better complexity. The theoretical results are verified in several tasks, and the proposed algorithm can consistently achieve prominent acceleration.

**Limitations And Societal Impact:**

Yes

**Main Review:**

Compared with the existing works, this paper provides the first non-asymptotic analysis of optimization algorithms for DRO with general smooth non-convex losses $\ell(x,\xi)$ and general $\psi$-divergence. In this setting, there are two major difficulties : (i) the DRO objective $\Psi(x)$ is non-convex and can become arbitrarily non-smooth, causing standard techniques in smooth non-convex optimization to fail to provide a good convergence guarantee; (ii) the noise of the stochastic gradient of $\Psi(x)$ can be arbitrarily large and unbounded even if the gradient of the inner loss $\ell(x,\xi)$ is assumed to have bounded variance. Therefore, the core technique of this paper is to exploit the specific structure of $\Psi(x)$, which shows that (i) the DRO objective satisfies a generalized smoothness condition [Zhang et al., 2020a,b] and (ii) the variance of the stochastic gradient can be bounded by the true gradient. This motivates the authors to adopt the special algorithm that combines gradient normalization and momentum techniques into SGD, by which both non-smoothness and unbounded noise can be tackled, finally resulting in an $O(\epsilon^{-4})$ complexity similar to standard smooth non-convex optimization.

**Time Spent Reviewing:**

12

---

> ### Author Response · Authors · 2021-08-09
> **Response to Reviewer u9sJ**
>
> We sincerely thank the reviewer for providing thoughtful review and positive feedback.

---

### Official Review · Reviewer_qgwb · 2021-07-17

**Rating:** 4
**Confidence:** 2

**Summary:**

This paper studies the distributionally robust non-convex optimization for non-convex objectives. Authors provide a theorem for the convergence rate based on alternative assumption.

**Ethics Review Area:**

["I don’t know"]

**Limitations And Societal Impact:**

See the above comment!

**Main Review:**

I would like to say that the paper is well-written and it is easy to follow; however, I have the following concerns:

(1) In general optimization theory it is known that bounded gradient assumption does not hold in practice and it grows with the size of input unless the updating rule involves projection to a certain set to keep the size of gradient bounded. Therefore, I think the provided example 3.1 (while being insightful) is a known fact in optimization theory.

(2) To deal with the issue (1) the reference [1] suggests the bounded variance assumption which is very similar to the assumption 3.2 in this paper. And using this assumption and careful choice of learning rate they can eliminate the effect of the growing norm of gradient. I believe this paper also follows the same approach and I found the results to be not surprising.

[1]: Bottou, L., Curtis, F. E., & Nocedal, J. (2018). Optimization methods for large-scale machine learning. Siam Review, 60(2), 223-311.

Given (1) and (2) I am struggling to support the acceptance of this paper, and while having an interesting direction and problem in hand, I think authors can improve the results further.

I also mention that I can be considered knowledgeable in optimization theory in general but I am totally new to the distributionally robust optimization area.

**Time Spent Reviewing:**

4 hours more or less

---

> ### Author Response · Authors · 2021-08-09
> **Response to Reviewer qgwb**
>
> We sincerely thank the reviewer for the detailed review. Below are our responses to the reviewer's concerns.
>
> **The difference between our approach and standard analysis.** We respectfully disagree with the reviewer's comment that our paper follows the same approach as [1]. In fact, the analysis in [1] heavily relies on the smoothness and bounded variance property of the objective function. Unfortunately, neither of the two properties hold in DRO setting, thus the approach in [1] can not be applied. We would like to stress that although we make assumptions (e.g. smoothness and bounded variance) on the inner loss, we are minimizing the DRO objective which does not inherit the nice properties of the inner loss, and the DRO objective is non-smooth and has unbounded gradient variance instead (see Lemmas 3.3 and 3.4).
>
> As a result, the proof of the main theorem (Theorem 3.5) is totally different from the proof in [1] and is far more challenging (please see the length of our proof). In the proof of [1], the objective function simply descends in every step (in expectation) even for vanilla SGD, which directly leads to $O(\epsilon^{-4})$ gradient complexity for finding an $\epsilon$-stationary point  with a constant learning rate $\eta=\Theta(\epsilon^2)$.
>
> In contrast, the algorithm we propose in this paper uses both adaptive step size (i.e. normalization) and momentum technique. Our main contribution is to show that this algorithm provably converges despite the non-smoothness and unbounded gradient variance of the DRO loss.
>
> **Implication of Example 3.1.** The main purpose of presenting Example 3.1 is to highlight the difference between DRO and stochastic optimization settings. The example shows that even if we make nice assumptions such as smoothness and bounded variance on the inner loss function, it is still possible that the DRO objective is non-smooth and the variance of the stochastic gradients is unbounded. As a result, conventional analysis in non-convex optimization (e.g. [1]) can not be applied in DRO setting.
>
> Above are some clarifications regarding the reviewer's concern and we sincerely hope that the reviewer can reconsider the score after reading our responses.
>
>
> [1] Bottou, L., Curtis, F. E., \& Nocedal, J. (2018). Optimization methods for large-scale machine learning. Siam Review, 60(2), 223-311.

---

> > ### Author Response · Authors · 2021-08-26
> > **Response to Reviewer qgwb**
> >
> > Hello reviewer qgwb, we would be grateful if you can confirm whether our response has addressed your concerns and please let us know if any questions remain. To recap our response.
> > - We highlight that our setting is different from the standard stochastic optimization setting, because we make assumption on the original loss and the goal is to minimize the DRO loss.
> > - As a result, the DRO loss is possibly non-smooth non-convex, and can have stochastic gradients with unbounded variance. This makes standard analysis of SGD inapplicable in our setting.
> > - Our main contribution is to provide an algorithm that provably overcomes the above challenges. Since the DRO loss does not inherit the properties of the original loss, our analysis is much more complicated than the standard approach.

---

### Official Review · Reviewer_uKhX · 2021-07-23

**Rating:** 8
**Confidence:** 3

**Summary:**

The authors propose an algorithm that they call mini-batch normalized gradient descent with momentum. This algorithm can find an \epislon-first-order stationary point under O(\epsilon^{-4}) gradient complexity, even though the objective function is non-convex and non-smooth and has unbounded gradient noise. Their work is limited to where all loss functions are G−Lipschitz and L−smooth in x and conjugate function of a divergence function is M−smooth. Their main algorithm goes like: get a mini-batch gradient estimator (using mini-batch to ensure convergence) and update momentum using that estimator and update their variable w using normalized momentum. They presented two numerical results: one classification, the other regression. They compared their algorithm with vanilla SGD regarding convergence speed.

**Main Review:**

This is a high quality paper that appears to give the first non-asymptotic convergence results for DRO problems with general smooth *non-convex* losses and general divergence functions. The paper is well-written, clear, and the results (to the best of my knowledge) appear to be sound and mathematically correct. The contribution is clearly significant due to the importance of non-convex optimization and DRO. I have no major criticisms and am happy to see this paper accepted.

**Time Spent Reviewing:**

2 hours

---

> ### Author Response · Authors · 2021-08-09
> **Response to Reviewer uKhX**
>
> We sincerely thank the reviewer for providing thoughtful review and positive feedback.

---

### Comment · Area_Chair_Se6T · 2021-09-13
**Comments and questions**

 Based on my own reading of the paper, I have the following comment about the algorithm and theory, plus a question regarding the experiment. Following general instruction from the program committee, I am posting them below to provide you an opportunity to respond.

**Comment on Scaling of $x$ vs. $\eta$.** The proposed algorithm uses the same step size of $x$ and $\eta$ gradient steps - this is arbitrary because the relative scaling of these variables depends on the parametrization of $x$. For example, running the algorithm on $\ell(x/c, \xi)$ instead of $\ell(x, \xi)$ would be equivalent to running the algorithm on the original $\ell$, but with different step sizes for $x$ and $\eta$. On a similar note, it is also arbitrary to require that $\Vert \nabla \mathcal{L}(x,\eta) \Vert \le \epsilon$, because $\nabla_x \mathcal{L}$ and $\nabla_\eta \mathcal{L}$ have different “units” and therefore should not be compared to the same quantity. The correct requirement should be  $\Vert \nabla_x \mathcal{L}(x,\eta) \Vert + G| \nabla_\eta \mathcal{L}(x,\eta) | \le \epsilon$. Beyond fixing the unit mismatch issue, this requirement also implies $\Vert\nabla \Psi(x)\Vert \le \epsilon$, which is arguably a more operationally-meaningful stationarity guarantee. Fixing this issue (by choosing a different step size of $\eta$ and modifying the convergence criterion) should also clean up your bounds: all expressions of the form $1+c G^2$ (which again depend on an arbitrary ratio between G and 1) should turn into $c’ G^2$ for some $c’$ close to $c$.

**Questions on experiments.**

1. Test performance matters more than training performance, yet as far as I could see the paper only reports training performance. How does the test performance compare across methods? Moreover, since the DRO objective is just a surrogate for out-of-distribution generalization, many DRO papers report generalization performance on more realistic shifted distributions. For the regression task, this could be MSE across different age bins, while for the classification task, it could be accuracy on the worst-performing class.

2. In the experiments, what would the graphs look like when plotting $\Psi$ rather than $\mathcal{L}$? Since the value of $\eta$ found in optimization has no bearing on the actual model predictions, I believe $\Psi$ is the more interesting quantity.

---

> ### Author Response · Authors · 2021-09-14
> **Response to Area Chair Se6T**
>
> We thank AC for the careful reading and valuable comments/questions. Below are our responses to these comments/questions.
>
> **Regarding the comment on scaling.** We have followed your advice by using the criterion $\Vert \nabla_x \mathcal{L}(x,\eta) \Vert + G| \nabla_\eta \mathcal{L}(x,\eta) | \le \epsilon$ instead of $\Vert \nabla \mathcal{L}(x,\eta) \Vert \le \epsilon$. In fact, the two criteria are equivalent up to a factor $G$, hence this will not change the $\mathcal{O}\left(\epsilon^{-4}\right)$ complexity, and our analysis still applies. By considering an auxiliary (scaled) function $\tilde{\mathcal L}(x,\eta)=\mathcal L(x,G\eta)$, we can obtain a complexity of $\mathcal{O}\left( \Delta G^2(L+\lambda^{-1}MG^2)(1+M^2\lambda^{-2}\sigma^2)\varepsilon^{-4}\right)$ using Theorem 3.5. This is equivalent to choosing different step sizes on $x$ and $\eta$ for optimizing $\mathcal{L}(x,\eta)$. We will make these modifications in the next version of this paper.
>
> **Regarding the experiments.** This paper focuses on the optimization of the penalty-based DRO objective.
> This objective has already been observed to be useful for improving test accuracy of worst-performing class in previous work [1].
> Since generalization is not our main focus, we did not include the results of test accuracy in the current version. For these results, our experiments show that the proposed algorithm can achieve better (worst-class) test accuracy compared with the standard optimization algorithm (SGD+momentum). For example, on unlabeled CIFAR-10 dataset, the worst-class accuracy using $\chi^2$-divergence and SGD+momentum is 44.80%, while the worst-class accuracy using normalized momentum rises to 49.75%. Note that we only sample 1425 training samples out of 5000 for this class (see Appendix E for details). We will include detailed results in an updated version of this paper.
>
> We calculate the objective $\Psi$ throughout training and find that the optimization performance in terms of $\Psi$ shows a similar trend compared with the curve w.r.t. $\mathcal{L}$ (Figure 1) in this paper. We will add plots of $\Psi$ for completeness.
>
>
>
> [1] Levy, D., Carmon, Y., Duchi, J. C., \& Sidford, A. (2020). Large-scale methods for distributionally robust optimization.

---

### Decision · Program_Chairs · 2021-09-27

**Decision:**

Accept (Poster)

**Comment:**

The paper provides convergence guarantees for a stochastic method on a family of non-convex DRO problems, getting around the non-trivial challenge of lacking global smoothness and variance bounds. The reviewers appreciated the quality of the writing, relevance of the topic, and the technical novelty of the results, and from my own reading of the paper I received a similar impression. When revising the paper, please take care to thoroughly address the comments provided by the reviewer and myself. In particular, it is crucial to provide test performance of the model you train, in order to show that the problems and hyperparameters for which you solve DRO are such that it really provides an improvement in robustness.